# Quantification of ongoing APOBEC3A activity in tumor cells by monitoring RNA editing at hotspots

Pégah Jalili [1], Danae Bowen [1], Adam Langenbucher[2], Shinho Park [1], Kevin Aguirre [1], Ryan B. Corcoran [2], Angela G. Fleischman [3], Michael S. Lawrence [2,4,5 ✉], Lee Zou [2,4 ✉] & Rémi Buisson [1,2 ✉]

APOBEC3A is a cytidine deaminase driving mutagenesis, DNA replication stress and DNA damage in cancer cells. While the APOBEC3A-induced vulnerability of cancers offers an opportunity for therapy, APOBEC3A protein and mRNA are difficult to quantify in tumors due to their low abundance. Here, we describe a quantitative and sensitive assay to measure the ongoing activity of APOBEC3A in tumors. Using hotspot RNA mutations identified from APOBEC3A-positive tumors and droplet digital PCR, we develop an assay to quantify the RNA-editing activity of APOBEC3A. This assay is superior to APOBEC3A protein- and mRNA-based assays in predicting the activity of APOBEC3A on DNA. Importantly, we demonstrate that the RNA mutation-based APOBEC3A assay is applicable to clinical samples from cancer patients. Our study presents a strategy to follow the dysregulation of APOBEC3A in tumors, providing opportunities to investigate the role of APOBEC3A in tumor evolution and to target the APOBEC3A-induced vulnerability in therapy.

[1] Department of Biological Chemistry, Center for Epigenetics and Metabolism, Chao Family Comprehensive Cancer Center, University of California Irvine, Irvine, CA, USA. [2] Massachusetts General Hospital Cancer Center, Harvard Medical School, Boston, MA, USA. [3] Department of Medicine, Chao Family Comprehensive Cancer Center, University of California Irvine, Irvine, CA, USA. [4] Department of Pathology, Massachusetts General Hospital, Harvard Medical School, Boston, MA, USA. [5] Broad Institute of Harvard and MIT, Cambridge, MA, USA. ✉email: mslawrence@mgh.harvard.edu; zou.lee@mgh.harvard.edu; rbuisson@uci.edu

Recent cancer genomics studies on tumors of a variety of cancer types have revealed important mechanisms driving tumor evolution and presented opportunities for cancer therapy. Among the 30+ mutational signatures identified to date, at least two are associated with APOBEC (Apolipoprotein B mRNA-editing enzyme catalytic polypeptide-like) proteins[1–3]. Members of the APOBEC3 family are capable of deaminating cytosine into uracil in DNA and/or RNA[4]. In particular, APO-BEC3A (A3A) and APOBEC3B (A3B), which target the TpC motif in single-stranded DNA (ssDNA)[4–7], are shown to be the main source of APOBEC-signature mutations in patients' tumors[1–3]. A3A and A3B are typically expressed after viral infection to defend cells against foreign DNA or RNA, but are not expressed in proliferating normal cells[4]. However, A3A and A3B expression is detected in ~50% of tumors[4,8,9]. In breast, lung, head & neck, esophageal, bladder, and cervical cancers, A3A and A3B are abnormally expressed, which is associated with high levels of APOBEC-signature mutations[2,4,8,10]. The mutagenesis induced by A3A and A3B in cancer cells may not only contribute to tumor evolution but also confer a vulnerability. The A3A and A3B activities in cancer cells induce DNA replication stress and DNA breaks[9–13], offering an opportunity to target the cancer cells harboring high levels of A3A/B activities by disrupting the DNA damage response (DDR)[10,14]. To exploit the A3A/B-induced vulnerability of cancer cells, it is critical to quantitatively measure the A3A/B activities in tumors. In this study, we investigated how to specifically quantify the activity of A3A in tumors.

While A3A and A3B have similar cytidine deaminase activities, their substrate specificities are not identical. Using yeast as a model to characterize A3A/B-mediated mutagenesis, the Gordenin laboratory identified differences at the substrate sites of A3A and A3B, with A3A preferring the YTCA motif and A3B favoring the RTCA motif (R is a purine and Y is a pyrimidine)[15]. In addition, we recently found that A3A but not A3B displays a preference for substrate sites in DNA stem-loops[8]. Importantly, recurrent APOBEC-signature mutations in different cancer patients are not randomly distributed in the genome but enriched in DNA stem-loops, suggesting that A3A is a major driver of mutations in tumors. Similar to its preference for DNA stem-loops, A3A also recognizes RNA stem-loops in a sequence-specific manner[16]. The distinct sequence and structural preferences of A3A and A3B have made it possible to distinguish the contributions of these enzymes to mutagenesis in tumors.

In addition to their differences in substrate specificity, A3A and A3B are also distinct in their catalytic activities. A3A is much more catalytically active than A3B, but A3B is typically expressed at higher levels than A3A in tumors. When overexpressed in cancer cells, A3A induces DNA replication stress, DNA double-strand breaks (DSBs), and cell cycle arrest in a catalytic activity-dependent manner[10–12,14]. Importantly, A3A-expressing cells become dependent on the Ataxia Telangiectasia and Rad3-related protein (ATR) checkpoint pathway to tolerate the DNA damage caused by A3A expression[10,14]. Inhibition of ATR[10,14] or its effector kinase Chk1[14] leads to increased cell death in an A3A-dependent manner, suggesting the potential use of ATR or Chk1 inhibitors to target tumors with high A3A activity. Furthermore, suppression of translesion synthesis (TLS) or base excision repair (BER), which are involved in tolerating and removing uracil in DNA, strongly sensitizes A3A-expressing cancer cells to ATR inhibitors[10]. We previously showed that depletion of endogenous A3A and A3B from certain cancer cell lines reduced their susceptibility to ATR inhibition, suggesting that both A3A and A3B may contribute to the vulnerability of tumors.

The quantification of A3A/B activities in tumors has been a challenge to date. Cell-extract-based in vitro assays have been developed to measure the activities of A3A/B. Although these in vitro assays are instrumental for understanding the biochemical properties of A3A/B, they are not easily applicable to tumor samples because of the large number of cells required. Furthermore, although given enough time A3A and A3B lay down clear mutational footprints in the genome, these footprints do not directly correlate with the currently ongoing activities of the enzymes. Interestingly, a recent study suggested that APOBEC mutagenesis occurs in an episodic manner in tumors[17], raising the possibility that persistent high A3A/B expression is not tolerated in cancer cells. Finally, due to the low expression of A3A, both A3A protein and A3A mRNA are difficult to quantify in limited numbers of cancer cells, making it particularly challenging to predict the levels of currently ongoing A3A activity in tumors.

In this study, we developed a strategy to quantify the ongoing activity of A3A in tumors. We found that A3A expression and A3A-mediated DNA mutagenesis in tumors, but not those of A3B, correlate with APOBEC-signature mutations in RNA stem-loops. These RNA mutations are not present in their DNA templates, suggesting that they are directly generated by A3A. Using hotspot APOBEC-signature mutations in RNA stem-loops identified from A3A-positive tumors and droplet digital PCR (ddPCR), we developed a quantitative and sensitive assay to measure the RNA-editing activity of A3A. Because of the labile, transient nature of RNA, the RNA-editing activity of A3A accurately reflects the currently ongoing activity of A3A. We demonstrate that the RNA mutation-based A3A assay is superior to APOBEC3A protein- and mRNA-based assays in predicting the currently ongoing A3A activity on DNA. Finally, we show that the RNA mutation-based A3A assay can be applied to clinical samples from cancer patients. Together, our results present a new strategy to follow the dysregulation of A3A in tumors, which will provide new opportunities to investigate the role of A3A in tumor evolution and to target A3A-induced vulnerabilities in cancer therapy.

## Results

**APOBEC3A mutations do not correlate with APOBEC3A expression.** Recent cancer genomics studies identified the APO-BEC signature as one of the most prevalent mutational signatures in tumors[1–3,18]. To identify the APOBEC+ tumors dominated by A3A or A3B, we sought to separate the contributions of A3A and A3B to the APOBEC-signature mutations (Fig. 1a, b). We analyzed the whole-genome sequencing (WGS) data of 1686 tumors of multiple cancer types from TCGA and other projects (Fig. 1a and Supplementary Table 1), and classified their mutations according to (1) frequency of APOBEC-signature mutations, (2) level of A3A character (the A3A-preferred YTC motif and DNA stem-loops), and (3) level of A3B character (the A3B-preferred RTC motif) (Fig. 1b). We clustered the tumors into three categories: tumors with high levels of A3A-signature mutations (A3A+), tumors with high levels of A3B-signature mutations (A3B+), and tumors with neither A3A- nor A3B-signature mutations (APOBEC-) (Fig. 1b). This analysis shows that the A3A signature (A3A+) is more prevalent than the A3B signature (A3B+) in cancers overall. However, the contributions of A3A and A3B vary in different cancer types. Bladder, cervical, head-and-neck, lung, endometrial, and thyroid cancers are dominated by A3A, whereas A3B is predominant in kidney cancer and sarcomas (Supplementary Fig. 1). Breast cancer is unique in including both many A3A-dominated as well as many A3B-dominated tumors (Supplementary Fig. 1).

Next, we tested whether A3A mRNA levels correlate with A3A-signature mutations in A3A+tumors (Fig. 1c, d). The correlation between A3A mRNA and A3A-signature mutations is poor (Fig. 1d), suggesting that A3A-signature mutations cannot predict A3A expression in tumors. This result indicates that many

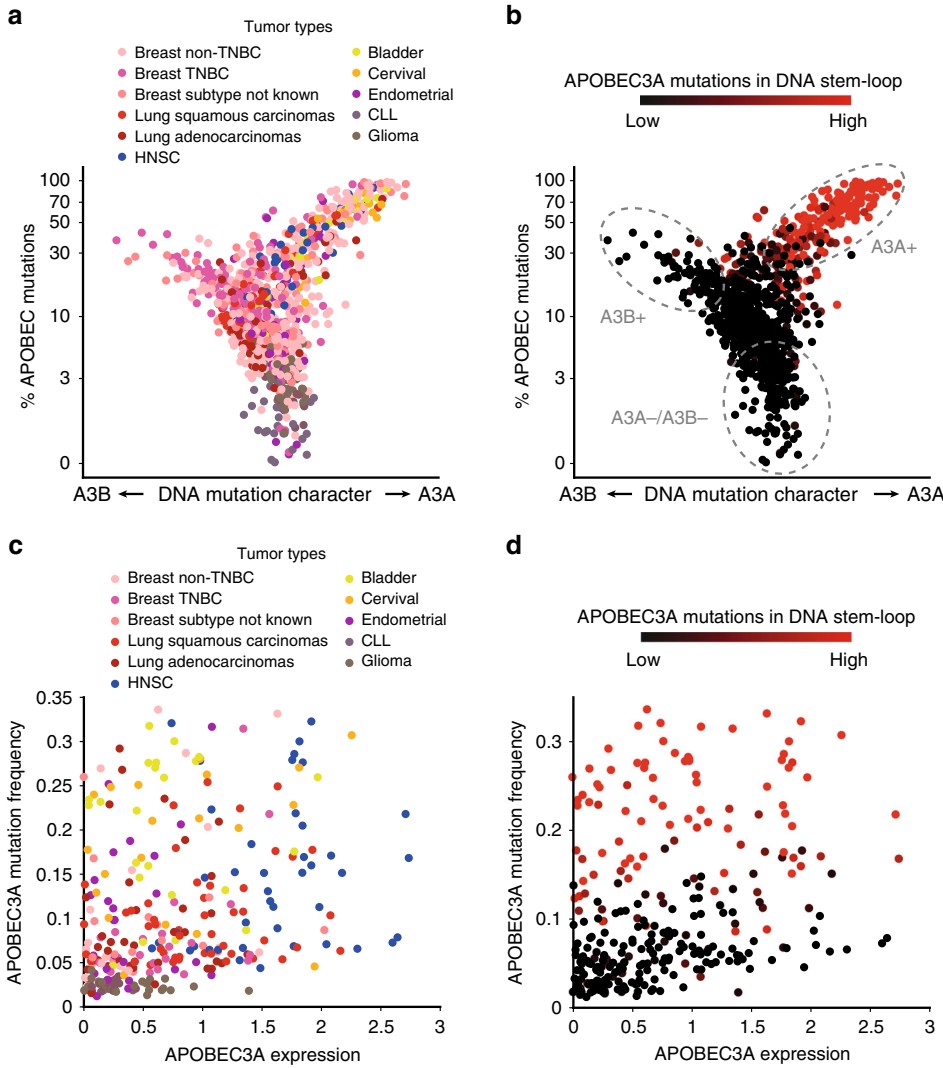

**Fig. 1 APOBEC3A and APOBEC3B mutation landscape in patients' tumors. a, b** Whole Genome Sequencing (WGS) of patient tumor samples (from TCGA and other projects, see Supplementary Table 1) were analyzed for their mutation frequency in the TpC motif. Each patient's tumor samples were plotted by their level of mutations in the TpC motif and their mutation frequency in RTC versus YTC sequences. **c, d** Patients' tumor samples were plotted according to their A3A expression level and mutation frequency in YTC sequences. Color-codes indicate patients' tumor types (**a, c**) or APOBEC3A mutation frequency in DNA stem-loops (**b, d**).

tumors carrying A3A-signature mutations no longer express A3A at the time of mRNA analysis, which is consistent with a recent report that APOBEC-signature mutations are generated in an episodic manner during tumorigenesis[17]. In addition, some tumors express A3A but do not have detectable levels of A3A-signature mutations (Fig. 1d), presumably because these tumors have not expressed A3A long enough to generate mutations that became sufficiently clonally expanded to be detected by bulk tumor sequencing. Therefore, we conclude that the levels of A3A expression and activity in tumors cannot be reliably inferred from the levels of A3A-signature mutations.

**In vitro APOBEC3A/B assay cannot predict APOBEC3A activity**. A cell-extract-based in vitro assay is commonly used to measure the APOBEC activity in cancer cells[5,8,9]. In this assay, a single-stranded DNA (ssDNA) oligo containing a single TpC site is incubated with cell extracts. When the C in the TpC site is deaminated by the APOBECs in cell extracts, the resulting U is cleaved by uracil DNA glycosylase (UNG), leading to an abasic site (AP site). The AP site in the ssDNA then undergoes site-specific

breakage in alkaline buffer at 95 °C, which can be visualized and quantified by electrophoresis (Supplementary Fig. 2a).

To test whether this in vitro APOBEC assay can be used to predict A3A activity, we selected a panel of cancer cell lines that express both A3A and A3B, A3B only, or neither A3A nor A3B (Fig. 2a, b). We did not identify any cell line expressing A3A only[10]. In this cell line panel, APOBEC activity was detected in all cell lines that express A3A and/or A3B (Fig. 2c, d). However, the APOBEC activity in these cell lines did not correlate with either A3A or A3B mRNA. Thus, the standard in vitro APOBEC assay cannot predict A3A in cancer cells.

We recently showed that A3A but not A3B prefers TpC sites in DNA stem-loops[8]. For example, a TpC site in the *NUP93* gene sits in a DNA stem-loop, and the DNA oligonucleotide containing this TpC site (NUP93) is a strong substrate of A3A in vitro (Fig. 2c). When the stem of NUP93 is disrupted, the resulting linear ssDNA oligo (polyA-TC) becomes a poor substrate of A3A (Fig. 2c). In contrast to A3A, A3B displays similar activities on NUP93 and polyA-TC. To test whether DNA stem-loop and linear substrates can help distinguish A3A and A3B activities, we tested the cell line panel with NUP93 and polyA-TC in vitro. Neither NUP93 nor

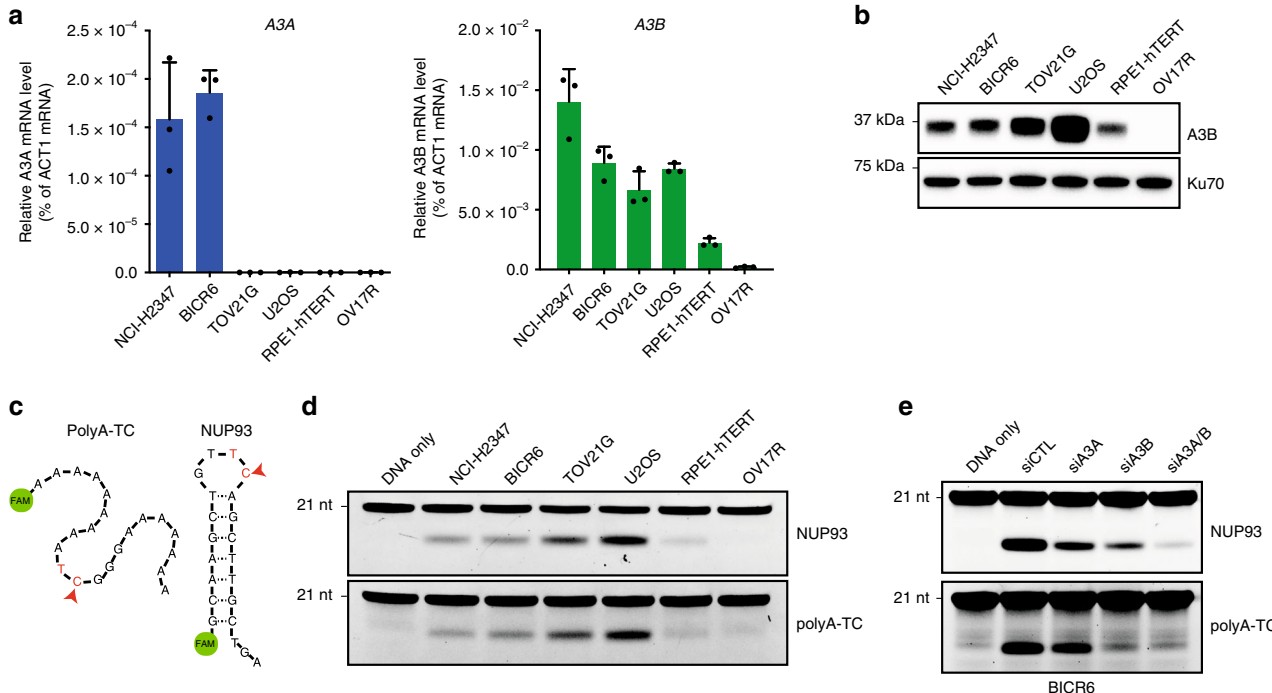

**Fig. 2 APOBEC3A expression does not correlate with APOBEC activity monitored by DNA deaminase activity assay. a** Analysis of A3A and A3B mRNA expression in a panel of cancer cell lines determined by RT-qPCR. Error bar: S.D. ($n = 3$). **b** A3B levels in the indicated cancer cell lines were analyzed by western Blotting. **c** Schematic representation of DNA oligonucleotides used in this study. polyA-TC: oligonucleotide without any secondary structure (single-stranded DNA). NUP93: natural occurring DNA hairpin in the gene *NUP93*. **d** Deamination activity on PolyA-TC and NUP93 substrates from indicated cell extracts (30 μg). **e** A3A and/or A3B knockdown in BICR6 cells. Deamination activity in BICR6 cell extracts (30 μg) was monitor after indicated knockdown on both PolyA-TC and NUP93 substrates. Source data are provided as a Source Data file.

polyA-TC elicited an activity that correlates with A3A level (Fig. 2d). BICR6 is a cell line that expresses both A3A and A3B (Fig. 2a, b). We used siRNAs to knock down A3A, A3B, or both A3A and A3B in BICR6 cells (Fig. 2e and Supplementary Fig. 2b, c). Depletion of A3A or A3B partially reduced the APOBEC activity on NUP93, whereas depletion of both A3A and A3B virtually abolished the activity. Because polyA-TC is a poor substrate for A3A, depletion of A3B reduced the activity on polyA-TC (71%) more than depletion of A3A (36%; Fig. 2e). These results suggest that both A3A and A3B contribute to the APOBEC activity detected in cell extracts. Furthermore, even when DNA stem-loop substrates are used, the in vitro APOBEC assay cannot predict A3A activity in cancer cells.

**APOBEC-signature mutations in RNA stem-loops in tumors.** A3A displays activity on not only ssDNA but also single-stranded RNA. An RNA-editing activity of A3A on UpC sites in stem-loops was detected in monocytes following inflammation or A3A overexpression[16,19]. However, whether the RNA-editing activity of A3A is present in cancer cells is still unknown. To investigate whether A3A modifies RNA in tumors, we compared A3A+ versus APOBEC- tumors and identified cytosines in the transcriptome that frequently acquire C->U mutations in RNA but are not mutated in the corresponding DNA from the same patient (Supplementary Figs. 3–6 and Supplementary Table 2). We reason that these RNA mutations are not generated by transcription of mutated DNA templates but instead are products of the RNA-editing activity of APOBECs. Sites undergoing APOBEC-dependent RNA editing showed an enrichment of the CAUC motif in stem-loops with 4-nt loops and with strong paired hairpins (Fig. 3a, b and Supplementary Fig. 7) as previously reported[16]. To further determine the structural specificity of the

APOBEC-generated RNA mutations, we restricted our analysis to stem-loops and compared loops differing by size and motif positioning (Fig. 3b). In A3A+ tumors but not A3A-/A3B- tumors, RNA mutations were identified in loops of 3, 4, and 5 nucleotides (Fig. 3b). In addition, the RNA mutations in A3A+ tumors were enriched at specific positions of the loops. For loops of 3, 4, and 5 nucleotides, the highest mutation frequency was observed when the U of the UpC motif is at the center of loops (Fig. 3b). Among all the APOBEC-signature RNA mutations in A3A+ tumors, the $DDOST^{558C>U}$ mutation is the most frequent (Fig. 3b, c and Supplementary Table 3). Close to 8% of the *DDOST* RNA is edited at position C558 in tumors displaying a strong A3A character. The average fraction of edited RNA for each RNA target is typically a few percent (Fig. 3b and Supplementary Table 3), however this can reach >30% in individual samples (Supplementary Fig. 5). Notably, the C558 of *DDOST* resides in a 4-nt loop formed by a CCAUCG motif (Fig. 3c), the optimal structural/sequence context for RNA mutagenesis in A3A+ tumors (Fig. 3a). Thus, the structural preferences of A3A for DNA and RNA stem-loops are similar, and A3A generates RNA hotspot mutations at optimal substrate sites in tumors.

To further investigate the contributions of A3A and A3B to RNA mutations in tumors, we selected patients' tumors that were analyzed by both DNA and RNA sequencing (Fig. 3d, e and Supplementary Fig. 8A, B). The APOBEC-signature RNA mutations are much more frequent in A3A+ tumors than in A3B+ tumors (Fig. 3d). In contrast to the DNA mutations (Fig. 1d), the RNA mutations strongly correlate with the mRNA levels of A3A but not A3B in tumors (Fig. 3e, Supplementary Fig. 8B). These results suggest that A3A but not A3B is the main cause of the APOBEC-signature RNA mutations in cancer cells, raising the possibility that RNA mutations are predictors of A3A activity in tumors.

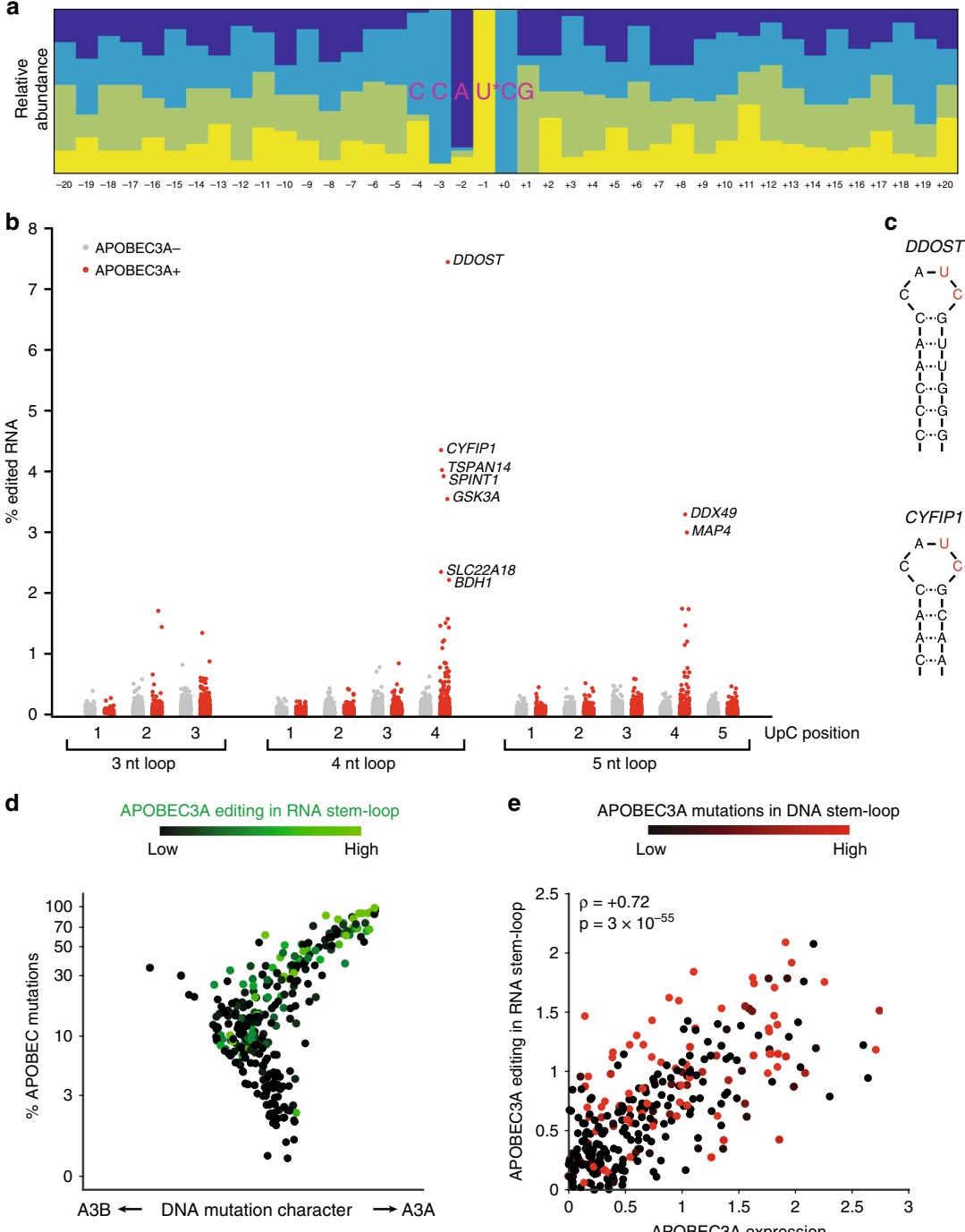

**Fig. 3 APOBEC3A promotes hotspot RNA mutations at specific stem-loop structures. a** Comparison of A3A-dominated tumors (APOBEC3A+) and tumors without APOBEC mutations or activity (APOBEC-) revealed sites undergoing frequent APOBEC-dependent C->U RNA editing. The top 50 most frequently edited sites were aggregated into a logogram showing relative frequencies of the bases at each position. **b** Classification of sites by hairpin characteristics revealed structural preferences of APOBEC-dependent editing. UpC sites exposed in a hairpin loop are mutated at higher frequencies. The position of the U in the loop affects mutability, with the highest RNA editing frequencies observed for C's positioned at the 3'-most position in a loop of four nucleotides. **c** Schematic representation of stem-loop structures in the genes *DDOST* and *CYFIP1*, the two most highly APOBEC-edited RNA sites in patient tumor samples. **d** RNA stem-loop editing levels detected in each patient were superimposed in green on the patients from Fig. 1b for whom RNA-sequencing data was available. **e** RNA stem-loop editing levels correlate strongly with the amount of A3A expression in patient tumors ($\rho = +0.72$, $p = 3 \times 10^{-55}$).

**An RNA mutation-based assay to measure APOBEC3A activity.** Next, we sought to develop a strategy to detect and quantify A3A-generated RNA mutations in tumors. Droplet digital PCR (ddPCR) is a quantitative and sensitive method to detect SNPs, rare mutations, copy number variations, and gene rearrangements. ddPCR is based on the partitioning of PCR reactions into small droplets following the Poisson distribution and resulting in partitions containing zero, one or more copies of the template sequence. After PCR amplification, the droplets are individually assessed for fluorescence, allowing an absolute quantification of the template

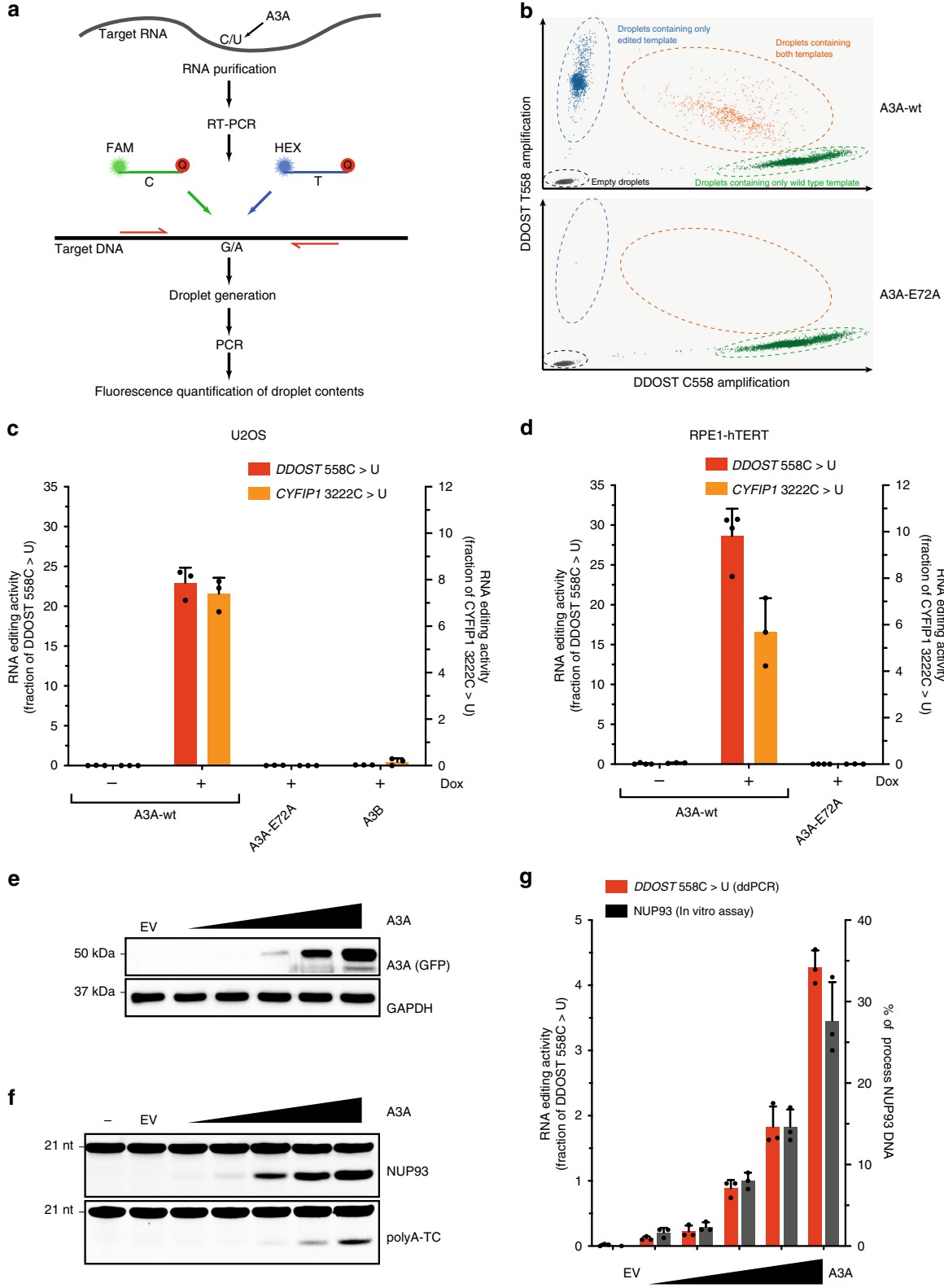

sequence. Importantly, two fluorescence probes can be used simultaneously to determine the proportions of two sequence variants in a population of molecules. Given these desirable features of ddPCR, we tested whether it can be used to quantify A3A-generated RNA mutations in cells.

We have previously generated U2OS-derived cell lines that inducibly express wild-type A3A (A3A[WT]) or a catalytically inactive

A3A mutant (A3A[E72A]) upon doxycycline (DOX) treatment (Supplementary Fig. 9A). Two days after A3A induction, total RNA is isolated, retrotranscribed to DNA, and then analyzed by ddPCR (Fig. 4a). To detect A3A-mediated RNA editing, we selected DDOST[558C>U] and CYPIF1[3222C>U], the two most frequent RNA hotspot mutations in A3A+ tumors, as targets for ddPCR (Fig. 3b, c). Both the DDOST[558C>U] and CYPIF1[3222C>U] mutations occur in

**Fig. 4 An RNA-based assay to detect APOBEC3A activity in cells. a** Schematic representation of the droplet digital PCR (ddPCR) strategy to detect RNA-editing by A3A. **b** Scatter plots of wild type (green) or edited (blue) *DDOST* amplified by ddPCR after expression of wild-type A3A or a catalytically inactive mutant A3A (A3A$^{E72A}$) in U2OS cells. **c, d** U2OS-derived or RPE1-hTERT-derived cell lines inducibly expressing A3A, A3A$^{E72A}$, or A3B were induced with doxycycline (DOX) or left uninduced. The level of edited *DDOST*$^{558C>U}$ and *CYFIP1*$^{3222C>U}$ in U2OS or RPE1-hTERT cells was quantified by ddPCR assay. Error bar: S.D. ($n \geq 3$). **e** HEK-293T cells were transfected with an increasing amount of vector expressing A3A-Flag/GFP. A3A protein levels in HEK-293T cell extracts were analyzed by western Blotting. E.V. empty vector. **f** Deamination activity on NUP93 and PolyA-TC substrates was measured following incubation with 0.2 μg or 1 μg of HEK-293T cell extracts, respectively, expressing an increasing level of A3A as shown in **e**. **g** Quantification of edited *DDOST*$^{558C>U}$ by ddPCR assay tracked closely with monitoring of cleavage of NUP93 DNA by Electrophoretic Mobility Shift Assay as shown in **f**. Error bar: S.D. ($n = 3$). For the ddPCR quantification, RNAs were purified from the same pool of HEK-293T cells shown in **e** and **f**. Source data are provided as a Source Data file.

RNA stem-loops that are optimal for A3A (Fig. 3c). We designed TaqMan probes labeled with FAM or HEX to recognize the wild-type (WT) or mutant sequence, respectively (Fig. 4a). The droplets displaying FAM (WT) and HEX (mutant) fluorescence after ddPCR were quantified and plotted in a two-dimensional scatter plot (Fig. 4b). The ratio between WT and mutant droplets was calculated to determine the frequency of the particular RNA mutation in cells. Induction of A3A$^{WT}$ but not A3A$^{E72A}$ in U2OS cells efficiently induced *DDOST*$^{558C>U}$ and *CYPIF1*$^{3222C>U}$ RNA mutations (Fig. 4b, c and Supplementary Fig. 9a). Similar results were also obtained in untransformed RPE1-hTERT cells (Fig. 4d). Notably, we did not detect any significant RNA editing activity in cells expressing only A3B (Fig. 4c and Supplementary Fig. 9B), suggesting that this activity is specifically attributable to A3A. Together, these results suggest that RNA mutation-based ddPCR can be used to detect the RNA-editing activity of A3A in cells.

Because A3A uses the same catalytic domain to deaminate ssDNA and RNA, it is expected that the RNA-editing activity of A3A correlates with its activity on ssDNA. To test whether the activities of A3A on ssDNA and RNA are tightly associated, we transiently expressed different amounts of A3A in HEK293T cells (Fig. 4e) and performed ddPCR and in vitro APOBEC assays in parallel (Fig. 4f, g). Because HEK293T cells do not express endogenous A3A/B, the only APOBEC activity detected in the in vitro assay comes from exogenous A3A (Fig. 4f). As the levels of A3A rose in HEK293T cells, both the RNA-editing activity detected by ddPCR and the DNA deamination activity detected by the in vitro assay increased accordingly. Importantly, the increase of RNA-editing activity was proportional to the increase of DNA deamination activity, showing that the two activities of A3A are tightly associated. Therefore, the ddPCR A3A RNA-editing assay can be used to predict the activity of A3A on DNA.

To test whether the ddPCR assay is sensitive enough to detect the activity of endogenous A3A in cancer cells, we selected a panel of cancer cell lines that express A3A and/or A3B at various levels (Fig. 5a). The levels of *DDOST*$^{558C>U}$ and *CYFIP1*$^{3222C>U}$ RNA mutations were analyzed with the ddPCR assay. To distinguish positive signals from background noise and establish a threshold for significant RNA-editing activity, we manually applied a cutoff of three HEX fluorescence-positive droplets[20] and 0.25 mutated *DDOST* copies per microliter (20,000 droplets were generated for each sample in the initial step of ddPCR). Significant RNA-editing activity was detected in NCI-H2347 and BICR6, two cell lines expressing A3A (Fig. 5b). In contrast, the signals detected in the other cell lines expressing only A3B or neither A3A nor A3B were below the cutoff for reliable RNA mutations (Fig. 5b), suggesting that they lack significant A3A activity. Moreover, knockdown of A3A abrogated RNA editing in BICR6 cells, confirming the specificity of the ddPCR assay for A3A activity (Fig. 5c and Supplementary Fig. 10A). These results further support the notion that A3A but not A3B promotes RNA hotspot mutations in cancer cells. Furthermore, the RNA mutation-based ddPCR assay is a

specific and sensitive method to quantify the RNA-editing activity of endogenous A3A in cancer cells.

**Superiority of RNA mutation-based APOBEC3A assay.** In addition to A3A RNA-editing activity, A3A protein and mRNA levels may also predict the A3A activity on DNA. We directly compared the ddPCR assay against the A3A protein- and mRNA-based assays for their abilities to monitor A3A expression/activity. We found that treatment of BICR6 cells with gemcitabine and interferon-α induced A3A expression (Fig. 6a–c and Supplementary Fig. 10B). This induction of A3A was detected by quantitative PCR (qPCR) of A3A mRNA, western blot of endogenous A3A protein, and ddPCR of A3A-generated RNA hotspot mutations. Thus, for a robust induction of A3A, all three methods work well. Notably, however, only qPCR and ddPCR, but not western blot, detected the baseline A3A expression/activity in untreated BICR6 cells (Fig. 6a–c), suggesting that qPCR and ddPCR are more sensitive than western blot. Similar to that in BICR6 cells, A3A was induced by gemcitabine and interferon-α in PC9 cells. While baseline expression of A3A was undetectable in PC9 cells (Supplementary Fig. 10C), the induction of A3A was clearly detected by both qPCR and ddPCR (Fig. 6d, e). However, in contrast to the robust induction in BIRC6 cells, the induction of A3A protein in PC9 cells was too weak to be detected by western blot (Fig. 6f). These results also support the notion that qPCR and ddPCR are more sensitive than western blot in detecting A3A expression/activity.

Given that APOBEC-signature mutations are likely generated in an episodic manner during tumor evolution[17], the expression of A3A in tumors may be transient. To test whether qPCR, western blot, and ddPCR can reliably monitor transient expression/activation of A3A, we treated BICR6 cells with gemcitabine and interferon-α, and then released them into drug-free medium. As detected by qPCR, A3A mRNA declined quickly after the release (Fig. 6a). However, western blot and ddPCR assays showed that A3A protein and its RNA-editing activity remained high for at least 3 days after the release (Fig. 6b, c). In PC9 cells, although A3A protein was not detected by western blot, qPCR, and ddPCR assays revealed that A3A mRNA declined more quickly than A3A activity after the release (Fig. 6d–f). Thus, in the context of transient A3A expression, A3A mRNA is not a reliable predictor of A3A protein and activity. This may be because A3A protein is significantly more stable than A3A mRNA. In this setting, ddPCR is more accurate than qPCR for monitoring transient A3A expression/activity.

**Detection of APOBEC3A activity in blood cancer samples.** Next, we tested whether the RNA mutation-based ddPCR assay can be applied to clinical samples from cancer patients. A recent study showed that A3A is highly expressed in AML (adult acute myeloid leukemia) cells[14]. To investigate whether A3A is commonly

expressed in blood cancers, we collected blood samples from 18 patients with AML or MPN (myeloproliferative neoplasms) (Supplementary Table 4). The amounts of RNA isolated from these samples varied significantly among patients. When we analyzed A3A mRNA using qPCR, the cycle threshold (Ct) values for the reference gene (actin) were high (Ct > 25) in some samples due to the low abundance of mRNA. Furthermore, the Ct values for the reference gene varied drastically (up to 7 cycles) among samples, making it unreliable to compare the A3A mRNA levels in different samples with qPCR. In contrast to qPCR, ddPCR is a method of direct quantification of DNA molecules of specific sequences,

(Fig. 7a). Notably, even in samples with very low levels of RNA, we detected significant A3A activity above the threshold of three HEX fluorescence-positive droplets and 0.25 mutated *DDOST* copies per microliter. Signals from patient samples UCI-642, UCI-632, UCI-463, 2015-004, and 2019-023 were below this threshold, suggesting that they are negative for A3A activity. To validate the quantification of A3A activity by ddPCR, we selected 4 samples with similar Ct values (Ct = 20 ± 1) for the reference gene, which allow us to compare the A3A mRNA levels in these samples by qPCR (Fig. 7b). Two of the samples had low or no detectable A3A mRNA and activity, whereas the other two had high levels of both A3A mRNA

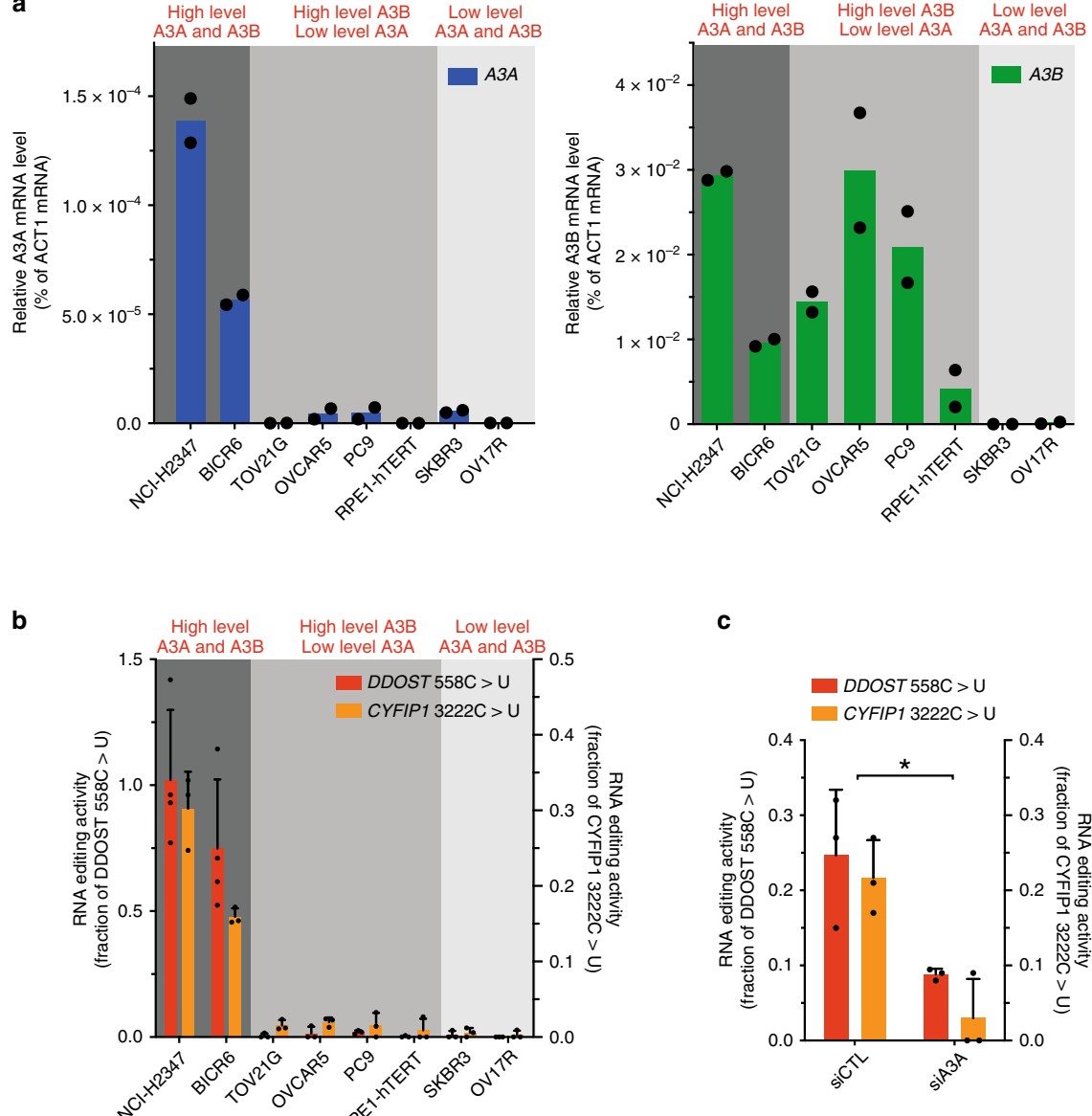

**Fig. 5 Correlation between APOBEC3A expression and RNA editing activity in a panel of cancer cell lines. a** Analysis of A3A and A3B mRNA expression in a panel of cancer cell lines determined by RT-qPCR. Error bar: S.D. (*n* = 2). **b** Quantification of *DDOST*[558C>U] and *CYFIP1*[3222C>U] levels in BICR6 cells by ddPCR assay after A3A knowndown. Error bar: S.D. (*n* ≥ 3). **c** Quantification of edited *DDOST*[558C>T] and *CYFIP1*[3222C>T] by ddPCR assay. Error bar: S.D. (*n* = 3). (**p* = 0.0352 for *DDOST* and *p* = 0.0111 for *CYFIP1* by two-tailed *t*-test) Source data are provided as a Source Data file.

making it possible to directly determine the copy numbers of wild-type and edited templates in each sample and the efficiency of editing. When we analyzed the RNA from the blood samples using the ddPCR A3A assay, a wide range of A3A activity was detected

and edited *DDOST*[558C>T] RNA (Fig. 7b, c). Thus, when A3A mRNA is quantifiable by qPCR in clinical samples, A3A mRNA largely correlates with A3A activity. However, the RNA mutation-based ddPCR assay can detect A3A activity even in samples that

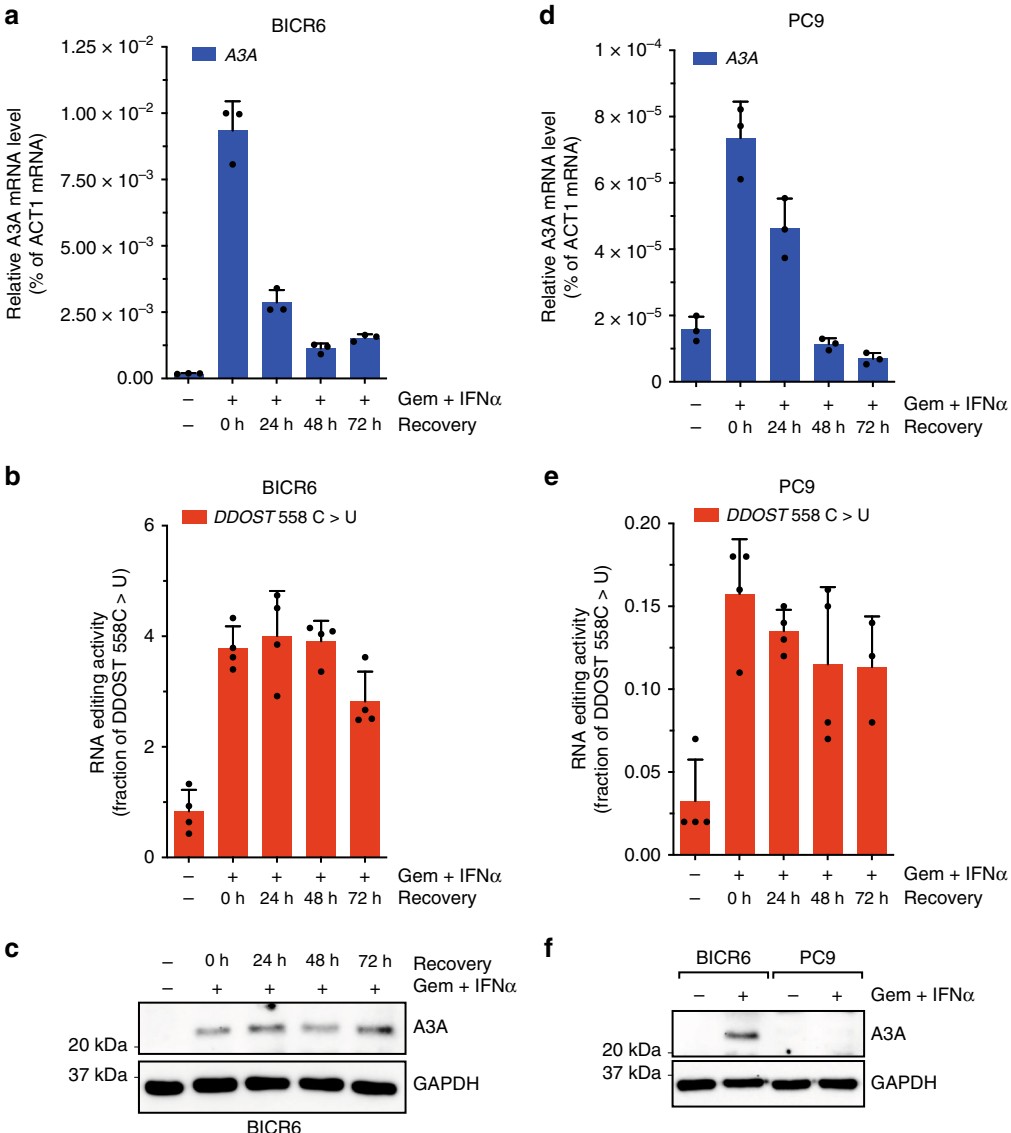

**Fig. 6 RNA mutation-based ddPCR assay is the most accurate method to monitor APOBEC3A level in cancer cells. a–f** Level of A3A was monitored by RT-qPCR, ddPCR, and western blot in BICR6 or PC9 cells following treatment with Gemcitabine (0.5 μM) and Interferon-αA/D (750 U ml⁻¹) for 48 h and release into drug-free media for 24 h, 48 h or 72 h. Error bar: S.D. ($n \geq 3$). Source data are provided as a Source Data file.

cannot be reliably analyzed by qPCR due to the low abundance of mRNA, suggesting that the ddPCR assay is a more sensitive method for quantifying A3A activity in clinical samples from cancer patients.

## Discussion

From recent cancer genomics studies, APOBEC proteins have emerged as key drivers of mutagenesis in a variety of cancers[4]. APOBEC-mediated mutagenesis in cancer cells may contribute to tumor evolution in several ways. The mutations generated by APOBECs may activate oncogenes and inactivate tumor suppressors, increasing the proliferative or survival advantages of cancer cells[4,18,21,22]. APOBEC-induced mutations may also promote tumor heterogeneity, increasing the ability of tumors to metastasize or develop resistance to therapy[17,21]. Furthermore, we and others recently showed that A3A/B expression can induce replication stress and DNA damage, generating genomic instability beyond mutations[10,14]. The genomic instability induced by A3A/B offers an opportunity for targeted therapy. For example, A3A expression in cancer cell lines confers sensitivity to ATR

inhibitors[10,14]. To understand the role of A3A in tumor evolution and to target the A3A-induced vulnerabilities of cancer cells, it is critical to develop a quantitative and sensitive assay for measuring A3A activity in tumors.

Using the distinct mutational signatures of A3A and A3B, we developed a computational strategy to identify the tumors dominated by A3A or A3B. Surprisingly, in the tumors dominated by A3A, the levels of A3A-signature mutations do not correlate well with A3A mRNA levels, suggesting that A3A mutational footprints cannot reliably predict currently ongoing A3A activity in tumors. Consistent with our finding, a recent study suggested that APOBEC-signature mutations are generated in an episodic manner in tumors[17]. It is possible that A3A is transiently activated by viral infection or other oncogenic stresses in cancer cells, leaving mutational footprints even after A3A is no longer expressed. It is also possible that persistent A3A expression is not well tolerated in tumors because of its robust activity and its ability to induce high levels of genomic instability, thereby selecting for tumor cells that turn off A3A after transient expression. Regardless of the mechanism underlying the separation of A3A footprints and

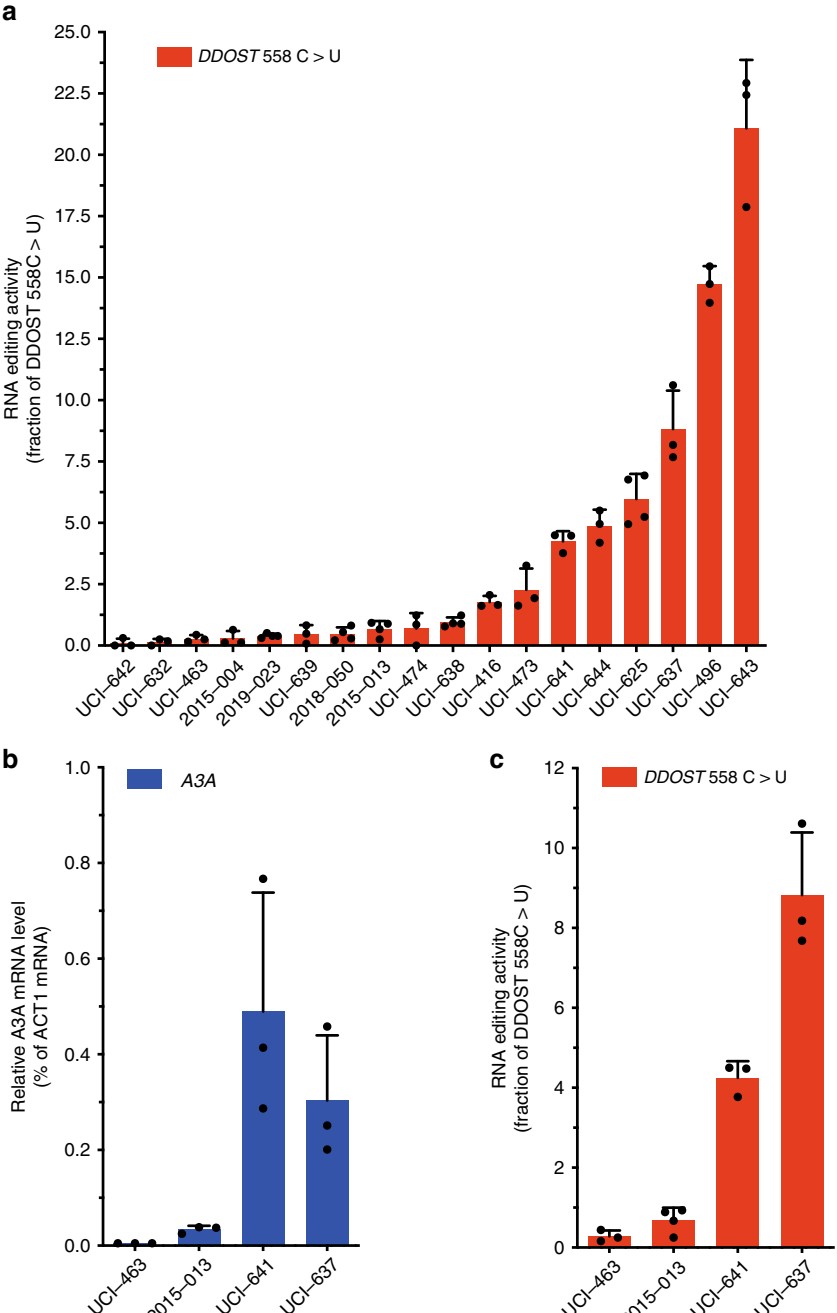

**Fig. 7 APOBEC3A activity in a panel blood tumors. a** Levels of edited $DDOST^{558C>U}$ were quantified using the ddPCR assay in a panel of patient blood tumor samples. For each sample, independent PCR reactions were performed to determine the level of edited $DDOST^{558C>U}$. Error bar: S.D. ($n \geq 3$) **b**, **c** Levels of A3A mRNA and edited $DDOST^{558C>U}$ were quantified by RT-qPCR and ddPCR respectively in the indicated patients' blood tumor samples. For each sample, three or more independent PCR reactions were performed to determine the level of A3A mRNA or $DDOST^{558C>U}$. Error bar: S.D. Edited $DDOST^{558C>U}$ level results showed in **c** were duplicated from the **a** of the corresponding patient samples. Source data are provided as a Source Data file.

expression, our findings suggest that the A3A mutational signature is not a reliable predictor of currently ongoing A3A activity in tumors.

A3A is known to modify not only DNA but also RNA in response to viral infection. In this study, we found that A3A-signature mutations are detected in RNA from tumors even in the absence of corresponding DNA mutations, suggesting that these RNA mutations are directly generated by A3A. The RNA mutations correlate with the expression of A3A but not A3B in tumors. Similar to the DNA mutations generated by A3A, the RNA mutations induced by A3A are enriched in stem-loops. However,

in RNA stem-loops, the sequence surrounding the UpC motif differs from the sequence surrounding the TpC motif in DNA stem-loops. For example, in RNA stem-loops with 4-nt loops, the UpC motif is most often preceded by A or U, whereas in DNA stem-loops, the TpC motif is usually preceded by C or T. We speculate that the difference of sequence preference is necessary to compensate for the structural difference between RNA and DNA backbones. The structure of A3A bound to single-stranded DNA showed several direct interactions between A3A and the sugar-phosphate backbone of DNA[6,23,24], suggesting that A3A may recognize RNA substrates in a slightly different way. Our finding

showed the activity of A3A on RNA closely correlates with its activity on DNA, suggesting that the current impact of A3A on DNA can be monitored by following A3A-mediated RNA editing. Nonetheless, the specific activities of A3A on equivalent DNA and RNA substrates may not be identical.

The functional significance of the RNA editing by A3A in cancer cells is still unclear. Because of the sequence conservation around the UpC motif (CAUC) in loops of four nucleotides, many A3A-generated RNA mutations in tumors convert one Isoleucine codon to another Isoleucine codon (AUC > AUU). In tumors, only a few missense mutations were detected in the RNA edited by A3A. Whether some missense RNA mutations generate gain-of-function or dominant-negative protein mutants remains to be investigated. The possible impacts of RNA mutations on non-coding RNAs also need to be explored. The fraction of edited RNA for each RNA target in tumors is typically 1–10% averaging across patients (Fig. 3b). It is possible that all or most of tumor cells have small fractions of the RNAs edited. Alternatively, small fractions of tumor cells may have high levels of edited RNAs. Our experiments using stable cell lines expressing A3A show that up to 30% of the *DDOST* RNA can be edited (Fig. 4c, d), suggesting that A3A has much more robust activity on certain RNAs than on their DNA templates. Moreover, edited *DDOST*[558C>T] RNA can reach >30% in individual tumors (Supplementary Fig. 5). It is tempting to speculate that A3A may mainly function as an RNA editor instead of a DNA mutator in certain oncogenic contexts, providing an alternative explanation of its dysregulation in cancers.

In contrast to mutations in DNA, RNA mutations cannot be inherited through genome duplication during cell divisions. Furthermore, due to the transient and labile nature of RNA, RNA mutations will disappear quickly after the responsible enzymes become inactive. Therefore, the RNA editing activity of A3A presents a unique opportunity to measure its currently ongoing activity. We took advantage of the hotspot editing sites in *DDOST* and *CYFIP1* mRNAs, as well as the highly sensitive ddPCR method, to develop a quantitative assay for the RNA editing activity of A3A. We confirmed that this assay is specific to A3A but not A3B. Importantly, this RNA mutation-based A3A assay is more sensitive than A3A protein- and mRNA-based assays in predicting A3A activity on DNA in cell lines, providing the most quantitative and sensitive assay to date for measuring currently ongoing A3A activity in cancer cells.

In this study, we demonstrate that the RNA mutation-based A3A assay can be applied to blood samples from blood cancer patients. Even in low-quantity samples that cannot be reliably analyzed by A3A RT-qPCR, the RNA mutation-based ddPCR assay detected A3A activity. It should be noted that ddPCR is already being used to detect pathogens, viruses, copy number variations, and rare mutations[25]. Furthermore, the Food and Drug Administration (FDA) has recently approved the use of ddPCR for monitoring the response of chronic myeloid leukemia (CML) patients to tyrosine kinase inhibitor treatment[26]. In future studies, it will be critical to test whether the RNA mutation-based A3A assay is sensitive enough to detect A3A activity in different types of tumor samples from cancer patients.

Cancer cells with significant A3A activity rely on DNA repair pathways and the ATR checkpoint to tolerate A3A-induced replication stress and genomic instability[10,14]. Inhibition of ATR in A3A-expressing cancer cells leads to replication catastrophe and cell death, and suppression of TLS or BER further enhances this sensitivity to ATR inhibitors[10]. This dependence of A3A-expressing cancer cells on the ATR checkpoint offers an opportunity for targeted therapy using ATR inhibitors. A number of clinical trials of ATR inhibitors are already underway[27,28]. The RNA mutation-based A3A assay that we developed may enable identification of tumors harboring significant A3A activity, helping to identify patients who may benefit from ATR inhibitor therapy. We envision that additional therapeutic strategies will be developed to exploit the A3A-induced vulnerability of cancer cells. The RNA mutation-based A3A assay may also be applied to facilitate these future therapies. In addition to guiding therapy, the RNA mutation-based A3A assay can be used to monitor the dysregulation of A3A during tumorigenesis. These studies may help us understand when, why and how A3A is activated and inactivated during tumor evolution. Defining the contexts and the windows of action for A3A may help address whether and how A3A contributes to tumor heterogeneity, metastasis, and drug resistance. We anticipate that the RNA mutation-based A3A assay will significantly advance our understanding of the function of A3A in tumorigenesis and allow us to exploit A3A-induced vulnerabilities in cancer therapy more effectively.

## Methods

**Computational APOBEC analyses.** Paired tumor-normal sequencing data from The Cancer Genome Atlas (TCGA) and other projects was analyzed to study molecular correlates of APOBEC enzyme activity at the DNA and RNA levels. Data from whole-exome DNA sequencing (WXS), whole-genome DNA sequencing (WGS), and RNA sequencing (RNA-Seq), was included in the analysis, all aligned to human genome build hg19. Detailed descriptions of the computational analysis are available in Supplementary methods.

**Plasmids.** APOBEC3A and APOBEC3B cDNA were synthesized by GenScript with a beta-globin intron and a Flag tag at C-terminus. The catalytically dead mutant APOBEC3A-E72A was constructed by site-directed mutagenesis. The plasmid expressing APOBEC3A-Flag, APOBEC3A-E72A-Flag, and APOBEC3B-Flag were generated by inserting the cDNA into pInducer20 or pDEST53 vectors using the Gateway Cloning System (Thermo Fisher Scientific).

**Cell culture.** U2OS-derived cell lines were generated by infecting U2OS cells with lentivirus expressing APOBEC3A, APOBEC3A-E72A, or APOBEC3B under a doxycycline-inducible promoter (pInducer20) and selected with G418 (400 μg mL$^{-1}$)[10]. U2OS derivative cells were maintained in DMEM supplemented with 10% FBS and 1% penicillin/streptomycin. For proteins expression, U2OS cells were incubated with doxycycline (200 ng mL$^{-1}$) for 48 h. U2OS, RPE1-hTERT, and HEK-293T cells were maintained in DMEM supplemented with 10% FBS and 1% penicillin/streptomycin. OVCAR5, PC9, and NCI-H2347 cells were maintained in RPMI-1640 GlutaMAX-I supplemented with 10% FBS, 1% penicillin/streptomycin, 1% glucose, and 1% sodium pyruvate. TOV21G, OV17R, and BICR6 cells were maintained in DMEM/F12 GlutaMAX-I supplemented with 10% FBS and 1% penicillin/streptomycin. SKBR3 was maintained in McCoy's 5A supplemented with 10% FBS and 1% penicillin/streptomycin. The cell lines above were purchased from either ATCC or Sigma-Aldrich.

**RNA interference.** siRNA transfections were performed by reverse transfection with Lipofectamine RNAiMax (Thermo Fisher Scientific). siRNAs were purchased from Thermo Fisher Scientific (Silencer® Select siRNA). Cells were treated with indicated drugs 16 h after siRNA transfection. The sequences of the siRNAs used in this study were:

siCTL: Catalog #4390846
siAPOBEC3A: CGACAGUACCAGACUCCAUtt
siAPOBEC3B: CCUCAGUACCACGCAGAAAtt

**Antibodies.** The antibodies used in this study were: Flag-M2 monoclonal antibody (Sigma-Aldrich #F1804)

GAPDH polyclonal antibody (EMD Millipore #ABS16), Flag polyclonal antibody (Sigma-Aldrich #F7425), APOBEC3B monoclonal antibody (Abcam #ab184990), APOBEC3A/B polyclonal antibody (Santa Cruz #sc-86289), and Ku70 monoclonal antibody (Gene Tex #GTX70271).

**Chemicals.** Gemcitabine was purchased from Selleckchem and dissolved in DMSO. Doxycycline and purified human Interferon-αA/D was purchased from Sigma-Aldrich.

**Cell extracts.** The APOBEC deamination assays were performed with cell extracts from the indicated cell lines[8,10]. Cells were lysed in 25 mM HEPES (pH 7.9), 10% glycerol, 150 mM NaCl, 0.5% Triton X-100, 1 mM EDTA, 1 mM MgCl$_2$, RNase A (0.2 μg ml$^{-1}$) and 1 mM ZnCl2, and protease inhibitors. Cell lysates were

sonicated, incubated for 30 min at 4 °C and then centrifuged 10 min at 16,000 × g at 4 °C. Protein concentration of the supernatant was determined by Bradford assay (Bio-Rad).

**DNA deaminase activity assay**. The deamination assays were performed as previously described[8,10]. Reactions (50 µL) containing 8 µL of a normalized amount of cell extracts (expressing A3A or A3B) were incubated at 37 °C during 1 h in a reaction buffer (42 µL) containing a DNA oligonucleotide (20 pmol of DNA oligonucleotide, 50 mM Tris (pH 7.5), 1.5 units of uracil DNA glycosylase (NEB), RNase A (0.1 µg mL$^{-1}$) and 10 mM EDTA). Then, 100 nM of NaOH was added to the reaction following by 40 min at 95 °C. Finally 50 µL of formamide was added to the reaction (50% final) and the reaction was incubated at 95 °C for 10 min following by 5 min at 4 °C. DNA cleavage was monitored on a 20% denaturing acrylamide gel (8 M urea, 1X TAE buffer) and run at 65 °C for 80 min at 150 V. DNA oligonucleotide probes were synthetized by Thermo Fisher Scientific. The sequences of DNA oligonucleotide probes used in this study are:

PolyA-TC: 5′-(6-FAM)-AAAAAAAAAT**C**GGGAAAAAAA-3′
NUP93: 5′-(6-FAM)-GCAAGCTGTT**C**AGCTTGCTGA-3′

**RT-qPCR**. Total RNA was extracted from cells using RNeasy Mini kit (Qiagen) according to the manufacturer's instructions. Following extraction, total RNA was reverse transcribed using the High Capacity cDNA Reverse Transcription Kit (Thermo Fisher Scientific). RT products were analyzed by real-time qPCR using SYBR Green (PowerUp SYBR Green Master Mix, Thermo Fisher Scientific) in a CFX Connect Real-Time PCR detection system (Bio-Rad). For each sample tested, the levels of indicated mRNA were normalized to the levels of Actin mRNA. The sequences of the PCR primers used in this study are:

Actin-forward: CCAACCGCGAGAAGATGA
Actin-reverse: 5′-CCAGAGGCGTACAGGGATAG
APOBEC3A-forward: GAGAAGGGACAAGCACATGG[9]
APOBEC3A-reverse: TGGATCCATCAAGTGTCTGG[9]
APOBEC3B-forward: GACCCTTTGGTCCTTCGAC[9]
APOBEC3B-reverse: GCACAGCCCCAGGAGAAG[9]

**Droplet digital PCR assay**. Purified RNAs were reverse transcribed using a High Capacity cDNA Reverse Transcription Kit (Thermo Fisher Scientific). In all, 20 ng (for DDOST$^{558C>T}$ amplification) or 40 ng (for CYPIF1$^{3222C>T}$ amplification) of cDNA, and indicated primers (2 µL) were added in the PCR reactions (ddPCR Supermix for Probes (No dUTP) mix from Bio-Rad) in a total of 25 µL. Then, 20 µL of the reaction mix was added to a DG8 cartridge (Bio-Rad), together with 70 µL Droplet Generation Oil for Probes (Bio-Rad) following by the generation of droplets using a QX200 Droplet Generator (Bio-Rad). Droplets were next transferred to a 96-well plate before to start the PCR reaction in thermal cycler (C1000 Touch Thermal Cycler, Bio-Rad) under the following conditions: 5 min at 95 °C, 40 cycles of 94 °C for 30 s, 53 °C for 1 min, and then 98 °C for 10 min (ramp rate: 2 °C s$^{-1}$). Droplets were analyzed with the QX200 Droplet Reader (Bio-Rad) for fluorescent measurement of fluorescein amidite (FAM) and hexachloro-fluorescein (HEX) probes. Gating was performed based on positive and negative DNA oligonucleotide controls. The ddPCR data were analyzed with QuantaSoft analysis software (Bio-Rad) to obtain fractional abundances of edited RNAs. Three or more biological replicates were analyzed for each sample. DDOST and CYFIP1 primers can be purchased from Bio-Rad (DDOST 558C #10031279, DDOST 558T #10031276, and CYFIP1 3222C/T #1863024).

**Primary human sample collection**. Peripheral blood or bone marrow was obtained from patients with hematologic malignancies. All participants gave their informed consent for the studies conducted in accordance with the Declaration of Helsinki. This study was approved by the Institutional Review Board of the University of California, Irvine. Mononuclear cells (MNCs) were isolated by density gradient using Ficoll-paque PLUS (GE Healthcare) and red blood cells were lysed by ammonium chloride potassium (ACK) lysing buffer. Cells were then immediately pelleted and stored at −80 °C until use. RNA was extracted from cells using RNeasy Mini kit (Qiagen) according to the manufacturer's instructions and analyses by qPCR or ddPCR.

**Statistics and reproducibility**. All western blots and DNA gels were repeated at least three times and representative images were shown in this paper.

**Reporting summary**. Further information on research design is available in the Nature Research Reporting Summary linked to this article.

## Data availability
The data that support this study are available from the corresponding authors upon reasonable request. Tumor sequencing data analyzed in this paper is publicly available from TCGA and can be downloaded from the Genomic Data Commons website (https://portal.gdc.cancer.gov/), accession number phs000178.v11.p8. The source data underlying

Figs. 2a, b, d, e, 4c–g, 5, 6, and 7, and Supplementary Figs. 2b, c, 4b, 7, 9, and 10 are provided as a Source Data file.

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

## Acknowledgements

We thank members of the Buisson, Lawrence, Corcoran, and Zou laboratories for helpful discussions and Melanie Oakes for advice on the droplet digital PCR. R.B. is supported by a NIH Pathway to Independence Award (CA212154), the California Breast Cancer Research Program, and the University of California Irvine Chao Family Comprehensive Cancer Center Anti Cancer Challenge. This work is supported by NIH grant CA218856 (L.Z.), MGH Cancer Center startup funds (M.S.L.), and an MPN Research Foundation Challenge grant (A.G.F.). This work was also made possible, in part, through access to the Genomics High Throughput Facility Shared Resource of the Cancer Center Support Grant (P30CA-062203) at the University of California, Irvine and NIH shared instrumentation grants 1S10RR025496-01, 1S10OD010794-01, and 1S10OD021718-01.

## Author contributions

R.B., L.Z., and M.S.L. designed the RNA mutation-based A3A assay, and R.B. established the assay. P.J., D.B., A.L., S.P., K.A., M.S.L., and R.B. performed all the experiments and analysis. R.B.C. helped with the droplet digital PCR. A.G.F. collected the blood patient samples used in this study. M.S.L., L.Z., and R.B. supervised the study and wrote the manuscript.

## Competing interests

L.Z. has received research funding from Calico. R.B.C. is a consultant/advisory board member for Amgen, Array Biopharma, Astex Pharmaceuticals, Avidity Biosciences, BMS, C4 Therapeutics, Chugai, Elicio, Fog Pharma, Fount Therapeutics, Genentech, Guardant Health, LOXO, Merrimack, N-of-one, Novartis, nRichDx, Revolution Medicines, Roche, Roivant, Shionogi, Shire, Spectrum Pharmaceuticals, Symphogen, Taiho, and Warp Drive Bio; holds equity in Avidity Biosciences, C4 Therapeutics, Fount Therapeutics, nRichDx, and Revolution Medicines; and has received research funding from Asana, AstraZeneca, and Sanofi. P.J., D.B., A.L., S.P., K.A., A.G.F., M.S.L., and R.B. declare no competing interest.
