## [Peer Review File · Nature Communications]

Reviewers' comments:

Reviewer #1 (Remarks to the Author):

Evidence from cancer genomic studies reveals mutational signatures in DNA that are associated with the APOBEC deaminase enzymes. This group recently reported that A3A shows a preference for substrates sites within DNA stem-loop structures. A3A has also been suggested to recognize stem loops in RNA. A3A is often expressed at low levels and is difficult to quantify accurately. This study describes an elegant and sensitive quantitative assay to measure A3A activity in tumor cells using an RNA substrate. This RNA signature assay has a number of distinct features to measure ongoing A3A activity. This assay is specific to A3A and not A3B and can be applied to clinical samples. It is a significance technical advance for the field that will be useful in analyzing cancer samples. The work is of high quality and well presented.

Figure 1 shows that expression levels and mutation signatures do not always correlate, which is reasonable and emphasizes the need for an activity assay. They detect A3A signatures in RNA stem-loops in tumors. Figure 2 show that the standard DNA deaminase assay is not sufficient. The data in Figure 3 from tumors suggest an RNA hotspot for A3A activity. The assay is developed and validated with an inducible cell line in Figure 4. Figures 5 and 6 show that it can be used for endogenous A3A in cancer cell lines. Figure 7 further validates in blood tumor samples and demonstrates that the activity assay is more sensitive than qPCR for low expression levels.

Specific suggestions:

- 1) How many tumor samples are being analyzed in Figure 3 and what is their origin?
- 2) The data in Figure 5A and 5B are the same as those in Figure 2A and 2B. This repetition should be avoided. The data for siA3A should be included in the main Figure 5 rather than the Supplementary Figure 5. The scales are different in Figure 5C and Supplementary 5C? The degree of knockdown should also be included.
- 3) Figure 6 could be combined with Figure 5.
- 4) Since the recognition of DNA and RNA substrates may be different, their dysregulation may be different. This could be commented on in Discussion.

Reviewer #2 (Remarks to the Author):

The manuscript "Quantification of ongoing APOBEC3A activity in tumor cells by monitoring RNA

editing at hotspots" addresses an important topic of quantification of protein, more specifically APOBEC3A, activity in cancers. Due to the difficulties with such quantification by established methods (e.g. western blot, qPCR gene expression), authors explored a new possibility to quantify the activity via the APOBEC-signature mutations in RNA, by implementing droplet digital PCR (ddPCR). The manuscript provides good evidence that such option is feasible, moreover, it is even outperforming current methods. This novel method would most probably be of interest to cancer community, as the same approach could most probably be applicable to other similar cases.

Manuscript is well structured and the whole story is easy to follow from initial experiments to confirm the idea by correlating mutations with expression, to final evaluation of the new assay. Nevertheless, below are some comments that should be addressed to improve the manuscript.

The results of WGS analysis presented in Fig.1 and Sup.Fig.1 show classification of data sets in different groups (APOBEC3A, APOBEC3B, APOBEC3A/B, APOBEC3B-). However, the background of this classification is not described in the methods, therefore authors should describe the principle or statistical model of this classification.

When describing the results of in vitro APOBEC assays to predict A3A activity, authors state there is no correlation of APOBEC activity with either A3A or A3B. Although this can be visually observed from the figures, it would be good if such statements would have some statistical background. Moreover, this results could perhaps be explained a bit from literature data showing poor correlation with gene expression and protein abundance (or even activity).

Authors state "Because polyA-TC is a poor substrate of A3A, depletion of A3B but not A# reduced the activity on polyA-TC significantly (Fig. 2E)." Is there any quantitative data based on blot analysis, or is the statement on significance based only on visual observation.

Figure 3B shows the amount of RNA editing and when referred to in the text, it is stated that in some cases RNA mutations were not identified. This is misleading, since Fig 3B shows edited RNA in all of the cases. probably authors need to decide on a threshold, based on which they have decided on the significant difference from background.

Strong correlation of RNA mutations with mRNA levels of A3A but not A3B in tumors was mentioned (with reference to Fig. 3E and Sup. Fig 3B), but there is no statistical value to support this statement.

In a brief description of ddPCR authors state that each of droplets contain "a single copy of the template sequence". This is not true, since the distribution of the templates in the droplets

follow Poisson distribution, meaning that individual droplets can harbor more than one template copy, especially in case of higher template concentration. Authors statement might be true in the case of RainDrop platform, but they used a Bio-Rad's system.

In the text it is written that there are two populations of droplets, displaying green (WT) and yellow (mutant) fluorescence. Please unify the colors, since the figures are showing green and blue colors for the two fluorescence channels.

There is a statement that authors "did not detect any RNA editing activity in cells expressing only A3B (Fig. 4C)", but the histogram shows at least something at the bottom. Authors should define the threshold for cases when the activity is definitely present. In the case of ddPCR it is possible that 1 or 2 positive droplets appear as a result, but e.g. based on no template controls you can see those also randomly appear there. Therefore, to have 95% confidence in the results, a threshold of 3 positive copies was proposed by some groups (e.g. 10.1021/acs.analchem.5b01208). The same thing can be observed in Fig. 5B, where there is some signal, however the text says it was not expressed.

It would be essential to propose such limit and also a limit of quantification. Moreover, the authors should address the relevance of measured A3A activity - what level could be considered as background and what would be the threshold that would call for medical action.

It is not clear whether some data on Fig 6AB is the same as on Sup.Fig 6B - in one way it seems like it is the same, but because the axis scales are different one cannot be sure.

Authors state that "In contrast to qPCR, ddPCR is a method of direct quantification (absolute quantification) and it does not require an internal reference." This is true, however in the case as in the manuscript, when they were comparing samples where RNA isolation was obviously not performing in the same way for all samples, the internal reference is essential, if one wants to make quantitative comparison between samples. Further authors state "even in samples with very low levels of RNA, we detected A#A activity that is statistically significant." There is no data that supports this statement. It would be good to see the correlation between the level of RNA and measured A3A activity.

Some figure descriptions are not appropriate. Figures should be self-standing, but in many cases there is not enough information in figure legend that would enable the reader to understand it (e.g. Fig 2, Fig 6, most of supplementary figures).

Legend of Figure 4 is missing some details (e.g. orange color description).

Figure 5 title starts with "Correlation", but the figure does not show any correlation.

Figure 2DE there is no label on first lanes - please add.

Figure 3A - although the legend says the figure shows relative abundance, I would propose adding y-axis label.

Figure 4 CD - the histograms are in red/green - I would suggest changing the color to blue/yellow.

Figure 4 CDG - the y-axes are different for different targets or assays, but they are scaled to present the results to appear as comparable. This is a bit misleading - it would perhaps be better to use only one log-scale.

Sup.Figure 5 - there is no data on statistical difference.

The number of tested replicates is not clearly described in the methods or results. In most of the cases it is written " $n > 3$ ". It is then also not clear, whether this means triplicate (3 wells) measurement of one sample in ddPCR or something else. For reliable measurements it is usually good to have more replicates, preferably even more dilutions.

The ddPCR reaction conditions are missing the information on primers and probe concentrations. It would also be better to include the sequences in the manuscript. How was the threshold for positive droplets determined in ddPCR?

David Dobnik

Reviewer #3 (Remarks to the Author):

Thank you for the opportunity to review the study of Jalili et al., where they described a novel quantification method of APOBEC3A activity in tumors.

The tool proposed is novel and can indeed have a strong impact in the field of epitranscriptomics / genomics. Aside from some strengthening of experimental data (particularly computational/informatics data, see below), I feel that there's a huge missed opportunity here to make a substantial conceptual advance. Because what the authors actually show that A3A editing can predict mutation, **in situ**. This is a big deal. Why?

Because while levels of editing here are considered "low", levels of mutation per genome per population doubling are almost always much lower (the authors can try to estimate this: a new mutation might be acquired in one of two alleles making the "rate" per genome minuscule; but a 7%-20% rate among transcripts, for well expressed genes, is not trivial in terms of in vivo

preferred catalytic substrate). This means that the transcript editing rate will always be much higher than the mutation rate on the cognate gene. Given that these proteins are anti-inflammatory, one can hypothesise that A3A's main enzymatic function in the cell as an editor first, with "in situ" mutation as collateral damage (of course proof of that rather than correlation, is a different story). Still, conceptually, this turns the whole idea of "off target" or "ectopic" mutation on its head. The authors might consider incorporating this even as an idea, in their study - this is completely up to them.

*note: "low" is not necessarily accurate; could be high editing within a heterogeneous cell population; also note that RNA editing itself imparts substantial heterogeneity - see Paz-Yaacov et al, Cell Reports 2015 among others; finally note that levels of the preeminent RNA editor ADAR1 are notoriously "bimodal" - and yet the "low editing" events can be just as important as the "high editing" ones - see Picardi et al, RNA 2017

Specific comments:

Abstract

- "APOBEC proteins ... in cancers.": Not all APOBEC proteins are deaminases and not all of them are driving mutagenesis in cancer (e.g. APOBEC2 and 4). The authors should rewrite this sentence so as to make this clear.

Introduction

- "Importantly, A3A-expressing cells become dependent on the ATR.": Define ATR as it is mentioned for the first time
- "These RNA mutations..." I would recommend that for base changes on the RNA, the authors should call these "RNA editing" events instead of "mutations", so that the standalone "mutation" term is not confused with regards to the nucleic acid it occurs (RNA or DNA). Alternatively they can be termed "base editing" events.

Results

APOBEC3A-signature mutations do not correlate with APOBEC3A expression in tumors

- Figure 1A-B: the visualisation is very well-done, however it is not clear what the x-axis measures. In the corresponding legend, the authors write "plotted by their level of mutations in the TpC motif and their mutation frequency in RTC versus YTC sequences"; but while the mutation frequency is clear (maybe rename the y axis accordingly), the authors should define more clearly what axis x measures, or in other words define what is meant by "level of mutations". The in-text explanation still does not address this fully.
- Figure 1C and 1D: please, rename the y axis titles so that it's clear that it's the mutation frequency.

APOBEC-signature mutations in RNA stem-loops in tumors

- “To investigate whether A3A... DNA from the same patient”: this is a very crucial step in the analysis. Several mutation calling pipelines fail to accurately report the mutations from WGS data. It is important that upon filtering the SNPs from the transcriptome SNVs for reporting the RNA editing events, the authors should make sure that the mutations they have called are real. That can be achieved with tools such as bam-readcount or IGV (manual spot checking). The authors should also provide the detailed bioinformatic analysis in the “APOBEC analysis in patient DNA-Seq and RNA-Seq data” under Materials & Methods. In particular they should report: how the data were mapped, to which reference human genome version, how DNA mutations were called (I recommend to take the union of at least 2 pipelines), how mutations were computationally validate, how RNA SNVs were called, how RNA editing was called, which sites were considered (statistical significance?), how the annotation was performed. And for all of the above all the crucial parameters that differ from the default settings should be reported.
- Additionally, the authors claim that the editing events or mutations they have called are in RNA or DNA stem loops. It seems that their top candidates (Fig 3B, 3C) are indeed mutated or edited in such loops, but there is barely any computational evidence throughout the manuscript for that. Authors should show that there’s an enrichment in such loops (perhaps fisher test compared to random positions) and also include this analysis in the annotation piece mentioned above.

Detection of APOBEC3A activity in blood cancer samples

- The 18 patients analysed from AML and MPN, are correlating indeed with the assay developed by the authors, demonstrating that the RNA editing activity detected is attributed to A3A activity. However, I recommend that the authors should analyse the cancer patients with low A3A expression as well where there they should see the reverse from what described in the cohort of 18 patients.

Point-by-point responses to reviewers' comments:

Reviewer #1 (Remarks to the Author):

Evidence from cancer genomic studies reveals mutational signatures in DNA that are associated with the APOBEC deaminase enzymes. This group recently reported that A3A shows a preference for substrates sites within DNA stem-loop structures. A3A has also been suggested to recognize stem loops in RNA. A3A is often expressed at low levels and is difficult to quantify accurately. This study describes an elegant and sensitive quantitative assay to measure A3A activity in tumor cells using an RNA substrate. This RNA signature assay has a number of distinct features to measure ongoing A3A activity. This assay is specific to A3A and not A3B and can be applied to clinical samples. It is a significance technical advance for the field that will be useful in analyzing cancer samples. The work is of high quality and well presented.

We thank the reviewer for his/her appreciation of the significance and quality of our work.

Figure 1 shows that expression levels and mutation signatures do not always correlate, which is reasonable and emphasizes the need for an activity assay. They detect A3A signatures in RNA stem-loops in tumors. Figure 2 show that the standard DNA deaminase assay is not sufficient. The data in Figure 3 from tumors suggest an RNA hotspot for A3A activity. The assay is developed and validated with an inducible cell line in Figure 4. Figures 5 and 6 show that it can be used for endogenous A3A in cancer cell lines. Figure 7 further validates in blood tumor samples and demonstrates that the activity assay is more sensitive than qPCR for low expression levels.

Specific suggestions:

1) How many tumor samples are being analyzed in Figure 3 and what is their origin?

In Figure 3, we focused on bladder, cervical, and head-and-neck cancer as diseases with common APOBEC involvement. We analyzed a cohort of 50 patients from TCGA (25 with low A3A levels and 25 with high A3A levels) and looked for UpC sites in the transcriptome that are hotspots for A3A-associated RNA editing. For comparison, we selected gastric cancer as a largely APOBEC-negative tumor type, and analyzed a cohort of 27 patients from TCGA with undetectable A3A signature. We have now added Supplementary Table 2 listing these patients, and we have added an expanded computational methods section with details of the analysis. Note, Figure 1 and Supplementary Figure 1 analyze a larger dataset including 1,686 patients spanning 27 tumor types. We have added Supplementary Table 1 listing these patients from TCGA and other published projects.

2) The data in Figure 5A and 5B are the same as those in Figure 2A and 2B. This repetition should be avoided. The data for siA3A should be included in the main Figure 5 rather than the Supplementary Figure 5. The scales are different in Figure 5C and Supplementary 5C? The degree of knockdown should also be included.

The analyses shown in Figure 5A and 5B were performed independently of those shown in Figure 2A and 2B. In addition, in Figure 5A, we expanded our panel of cell lines by adding OVCAR5, PC9, and SKBR3. It was important to show the levels of A3A and A3B in all the cell lines in Figure 5A in order to compare them directly and to correlate the levels of A3A expression and RNA editing activity.

Following the reviewer's suggestion, we have moved the siA3A data from Supplementary 5 to Figure 5. Please note that the experiments in Fig. 5B (not 5C) and Supplementary Fig. 5 (not 5C) are different. Only the samples in Supplementary Fig. 5 were transfected with siRNAs. We noted that the efficiency of RNA editing was reduced by transfection, even when Ctrl siRNA was used. This is why the scales of Fig. 5 and Supplementary Fig. 5 are different.

3) Figure 6 could be combined with Figure 5.

We appreciate the reviewer's suggestion. However, we think that Fig. 5 and Fig. 6 each makes a distinct point. Fig. 5 shows that RNA editing activity only correlates with A3A but not A3B expression in a panel of cancer cell lines. Fig. 6 shows that the ddPCR assay for RNA editing activity is more sensitive than RT-qPCR and Western blot in detecting transient changes of A3A activity. We hope the reviewer would agree with us that Figures 5 and 6 are logically organized.

4) Since the recognition of DNA and RNA substrates may be different, their dysregulation may be different. This could be commented on in Discussion.

Our data actually argue that the activity of A3A on RNA accurately reflects its activity on DNA. In Fig. 4E-G, we show that the activity of A3A on an efficient DNA substrate (NUP93) closely correlates with the A3A activity on the optimal RNA substrate (DDOST). Thus, when A3A is dysregulated in cancer cells, its activity is similarly reflected by DNA and RNA substrates. Of course, A3A may recognize DNA and RNA substrates in slightly different ways due to the differences in the sugar backbones. We do not wish to suggest that A3A has the same activity on equivalent DNA and RNA substrates. We mention this in the revised discussion.

Reviewer #2 (Remarks to the Author):

The manuscript "Quantification of ongoing APOBEC3A activity in tumor cells by monitoring RNA editing at hotspots" addresses an important topic of quantification of protein, more specifically APOBEC3A, activity in cancers. Due to the difficulties with such quantification by established methods (e.g. western blot, qPCR gene expression), authors explored a new possibility to quantify the activity via the APOBEC-signature mutations in RNA, by implementing droplet digital PCR (ddPCR). The manuscript provides good evidence that such option is feasible, moreover, it is even outperforming current methods. This novel method would most probably be of interest to cancer community, as the same approach could most probably be applicable to other similar cases.

Manuscript is well structured and the whole story is easy to follow from initial experiments to confirm the idea by correlating mutations with expression, to final evaluation of the new assay. Nevertheless, below are some comments that should be addressed to improve the manuscript.

We thank the reviewer for his/her appreciation of the significance and quality of our work.

The results of WGS analysis presented in Fig.1 and Sup.Fig.1 show classification of data sets in different groups (APOBEC3A, APOBEC3B, APOBEC3A/B, APOBEC3B-).

However, the background of this classification is not described in the methods, therefore authors should describe the principle or statistical model of this classification.

We thank the reviewer for raising this important point. The A3A and A3B mutation signatures are very similar, both targeting TpC sites in the genome, and so it is important to explain how we are able to distinguish them. The classification of the A3A vs. A3B mutational signatures is based on the published observation (Chan et al. Nature Genetics 2015 PMID 26258849) that A3A and A3B have distinct preferences for the nucleotide at the "minus-2" position (i.e. just before the TpC site). A3A preferentially targets sites in which the minus-2 position is a pyrimidine (C or T), whereas A3B prefers a purine (A or G) at that position. The Chan paper showed that this preference is observed in yeast models and in human cancer samples. We previously (Buisson et al. Science 2019) used this preference to distinguish A3A from A3B in APOBEC mutation signatures in TCGA samples sequenced by WGS. We have used the same procedure here, explaining how this "minus-2" preference allows us to distinguish tumors dominated by A3A mutagenesis, vs. those dominated by A3B mutagenesis. We show that breast cancer is unique among the diseases we studied, in that it includes cases of *both* types: individual patients dominated by A3B mutagenesis, as well as individual patients dominated by A3A mutagenesis. We have added additional detail to the computational methods section explaining exactly how we measured A3A vs. A3B mutational character.

When describing the results of in vitro APOBEC assays to predict A3A activity, authors state there is no correlation of APOBEC activity with either A3A or A3B. Although this can be visually observed from the figures, it would be good if such statements would have some statistical background. Moreover, this results could perhaps be explained a bit from literature data showing poor correlation with gene expression and protein abundance (or even activity).

We thank the reviewer for his/her suggestion. However, we are hesitant to compare a regression between two continuous variables as in the case for A3B expression and APOBEC activity compared to a correlation between a single continuous variable (APOBEC activity) versus a binary variable (A3A expression). Moreover, because we monitored mRNA level, protein level, and enzymatic activity level for A3A and A3B using different assays, it is difficult to generate accurate statistical values to compare results across panels from different assays. Our quantification of APOBEC activity in U2OS and TOV21G cells (which express only A3B) showed 2.6 and 1.7 times more APOBEC activity respectively than in BICR6 or NCI-H2347 cells (which express both A3A and A3B). This result strongly demonstrates that monitoring APOBEC activity using this in vitro assay cannot predict endogenous A3A expression in cancer cells.

To our knowledge, it has not been explained yet why A3B mRNA level does not always correlate with A3B protein level. Protein levels are often regulated by posttranslational modifications that can enhance or decrease protein stability in cells. It will be interesting in the future to study how A3B is regulated in cancer cells by such mechanisms.

Authors state "Because polyA-TC is a poor substrate of A3A, depletion of A3B but not A3A reduced the activity on polyA-TC significantly (Fig. 2E)." Is there any quantitative data based on blot analysis, or is the statement on significance based only on visual observation.

This statement is based on quantification of the cleaved DNA products in Fig. 2E. The quantified result shows that siA3B decreases the amount of cleaved polyA-TC DNA by

71%, whereas siA3A only decreases it by 36%. To address the reviewer's concern, we have now modified our statement as follows: "Because polyA-TC is a poor substrate for A3A, depletion of A3B reduced the activity on polyA-TC (71%) more than depletion of A3A (36%). (Fig. 2E)."

Figure 3B shows the amount of RNA editing and when referred to in the text, it is stated that in some cases RNA mutations were not identified. This is misleading, since Fig 3B shows edited RNA in all of the cases. probably authors need to decide on a threshold, based on which they have decided on the significant difference from background.

We apologize for the confusion. In the text, we actually stated that we specifically looked for the RNA mutations that do not have the corresponding DNA mutations detected. This strategy allowed us to distinguish the RNA mutations caused by A3A directly from those transcribed from mutated DNA templates. We did not state that RNA mutations were not detected at all in some tumors. We agree with the reviewer that a threshold is needed if one wants to separate tumors with high and low levels of RNA mutations.

Strong correlation of RNA mutations with mRNA levels of A3A but not A3B in tumors was mentioned (with reference to Fig. 3E and Sup. Fig 3B), but there is no statistical value to support this statement.

The statistical values were indicated in the figure legend ($\rho = +0.72$, $p = 3 \times 10^{-55}$). We have now included these numbers in Fig. 3E to improve clarity.

In a brief description of ddPCR authors state that each of droplets contain "a single copy of the template sequence". This is not true, since the distribution of the templates in the droplets follow Poisson distribution, meaning that individual droplets can harbor more than one template copy, especially in case of higher template concentration. Authors statement might be true in the case of RainDrop platform, but they used a Bio-Rad's system.

We thank the reviewer for pointing out this inaccurate statement. Indeed a small population of the droplets harbors more than one template as described in Figure 4B. Using the Poisson distribution, we have now been able to account for this minor population in our quantification. We have now modified this statement as follows: "ddPCR is based on the partitioning of PCR reactions into small droplets, most of which contain a single copy of the template sequence".

In the text it is written that there are two populations of droplets, displaying green (WT) and yellow (mutant) fluorescence. Please unify the colors, since the figures are showing green and blue colors for the two fluorescence channels.

We used green and blue colors in the graph shown in Figure 4A-B because this is the color code imposed by Bio-Rad analysis software. The same color code has been used in many other publications showing results obtained from ddPCR analysis using Bio-Rad equipment. For consistency with other published work, we decided to keep a similar color code in our figure. In order to avoid confusion for the reader, we now have modified our statement by: "The droplets displaying FAM (WT) and HEX (mutant) fluorescence after ddPCR were quantified and plotted in a two-dimensional scatter plot (Fig. 4B)".

There is a statement that authors "did not detect any RNA editing activity in cells expressing only A3B (Fig. 4C)", but the histogram shows at least something at the bottom. Authors should define the threshold for cases when the activity is definitely

present. In the case of ddPCR it is possible that 1 or 2 positive droplets appear as a result, but e.g. based on no template controls you can see those also randomly appear there. Therefore, to have 95% confidence in the results, a threshold of 3 positive copies was proposed by some groups (e.g. 10.1021/acs.analchem.5b01208). The same thing can be observed in Fig. 5B, where there is some signal, however the text says it was not expressed.

It would be essential to propose such limit and also a limit of quantification. Moreover, the authors should address the relevance of measured A3A activity - what level could be considered as background and what would be the threshold that would call for medical action.

We thank the reviewer for this important suggestion. Indeed, we often detected 1 to 2 HEX-positive droplets in samples without any A3A expression. For example, in a PC9 sample, we detected 1 HEX-positive droplet positive for the mutated sequence and 8,421 droplets positive for the wild-type sequence. A similar observation was made using a wild-type control synthetic DNA oligo as a template, suggesting that we reached the sensitivity limit of the ddPCR assay. For our samples, we only consider those showing three or more HEX fluorescence-positive droplets and 0.25 mutated DDOST copies per microliter as positive for A3A activity (20,000 droplets were generated for each sample in the initial step of the ddPCR). We used copies per microliter as suggested by Bio-Rad to account for the difference in droplet numbers between samples (number of HEX positive droplets) / (number of droplets X volume of one droplet). We now have modified our statement as follows: "did not detect any significant RNA editing activity in cells expressing only A3B (Fig. 4C)". In addition, we added the following statements as suggested by the reviewer: "To distinguish positive signals from background noise and establish a threshold for significant RNA-editing activity, we applied a cutoff of three HEX fluorescence-positive droplets²³ and 0.25 mutated DDOST copies per microliter (20,000 droplets were generated for each sample in the initial step of the ddPCR)." / "the signals detected in the other cell lines expressing only A3B or neither A3A nor A3B were below the cutoff for reliable RNA mutations."

It is not clear whether some data on Fig 6AB is the same as on Sup. Fig 6B - in one way it seems like it is the same, but because the axis scales are different one cannot be sure.

Yes, the data in Figure 6C are identical to the one from the same patient samples shown in Figure 6A. Because of the limited amount of samples collected from patients, we were limited in the number of assays we could perform on each sample. The goal of panels B and C of Figure 6 was to focus on four patient samples from panel A in which we obtained reliable RT-qPCR data, and to demonstrate that the levels of RNA editing in DDOST correlate with the levels of A3A mRNA in these four patient samples. We have now indicated in the figure legend that the data in panel C were duplicated from the corresponding patient samples showed in panel A.

Authors state that "In contrast to qPCR, ddPCR is a method of direct quantification (absolute quantification) and it does not require an internal reference." This is true, however in the case as in the manuscript, when they were comparing samples where RNA isolation was obviously not performing in the same way for all samples, the internal reference is essential, if one wants to make quantitative comparison between samples. Further authors state "even in samples with very low levels of RNA, we detected A#A activity that is statistically significant." There is no data that supports this statement. It would be good to see the correlation between the level of RNA and measured A3A activity.

In the ddPCR assay, the fraction of a particular RNA mutation was determined from the signals detected in all individual droplets (Fig. 4B). Therefore, we consider ddPCR a “self-normalized” measurement of the frequency of RNA mutation, and the internal reference for each sample is all the droplets in the same sample. When different RNA samples are compared, they may generate different numbers of droplets due to variations in the amounts of RNA analyzed. However, this won’t affect the ability of ddPCR to quantify the fraction of RNA mutation in each sample. For example, imagine that one RNA sample generates 100 droplets and 10 have the RNA mutation. Another sample generates 50 droplets and 10 have the RNA mutation. These results would suggest that the frequency of RNA mutation is 2 times higher in the second sample than in the first sample, even though the amounts of RNA are different in the two samples. It is important to note that the frequencies of RNA mutations in individual samples, rather than the absolute levels of RNA mutations in these samples, reflect A3A activity. Therefore, ddPCR is a “self-normalized” assay to quantify and compare A3A activity even among samples with different amounts of RNA. We have now modified our statement as follows: “even in samples with very low levels of RNA, we detected more than 3 HEX fluorescence positive droplets per sample, suggesting the presence of significant A3A activity.”

Some figure descriptions are not appropriate. Figures should be self-standing, but in many cases there is not enough information in figure legend that would enable the reader to understand it (e.g. Fig 2, Fig 6, most of supplementary figures).

We have now expended the figure descriptions in the figure legends.

Legend of Figure 4 is missing some details (e.g. orange color description).

Color descriptions used in panels of Figure 4 are indicated directly in the figure. We don’t think it was necessary to duplicate the information.

Figure 5 title starts with "Correlation", but the figure does not show any correlation.

We used the term “correlation” to convey the fact that APOBEC3A expression levels follow the RNA editing levels, indicating that APOBEC3A expression influences RNA editing level. These two molecular readouts correlate with each other.

Figure 2DE there is no label on first lanes - please add.

We have now added a label “DNA only” on the first lanes.

Figure 3A - although the legend says the figure shows relative abundance, I would propose adding y-axis label.

Thanks for this point. We have added a y-axis "relative abundance" label to Figure 3A.

Figure 4 CD - the histograms are in red/green - I would suggest changing the color to blue/yellow.

We have now modified the colors as suggested.

Figure 4 CDG - the y-axes are different for different targets or assays, but they are scaled to present the results to appear as comparable. This is a bit misleading - it would perhaps be better to use only one log-scale.

We have now modified the y-axes to make them comparable between panels.

Sup.Figure 5 - there is no data on statistical difference.

We have now added an analysis of the statistical difference. Both DDOST and CYFIP1 RNA editing activity decreased significantly upon A3A knockdown.

The number of tested replicates is not clearly described in the methods or results. In most of the cases it is written "n>3". It is then also not clear, whether this means triplicate (3 wells) measurement of one sample in ddPCR or something else. For reliable measurements it is usually good to have more replicates, preferably even more dilutions.

Three or more biological replicates have been performed for each ddPCR analysis. We have now clarified this in a statement in the methods section.

The ddPCR reaction conditions are missing the information on primers and probe concentrations. It would also be better to include the sequences in the manuscript. How was the threshold for positive droplets determined in ddPCR.

The ddPCR primers used in this study were designed by Bio-Rad upon our specific request. Bio-Rad uses a proprietary method to generate primers in order to identify a set of primers that can differentiate sequences with a single point mutation difference. Unfortunately, Bio-Rad did not disclose the sequences of the primers they designed for us. However, the primer can be ordered by anyone using the unique catalog number generated by Bio-Rad for these specific sets of primers. We have now included these catalog numbers for both the DDOST and CYPF11 primers in the methods section.

Reviewer #3 (Remarks to the Author):

Thank you for the opportunity to review the study of Jalili et al., where they described a novel quantification method of APOBEC3A activity in tumors. The tool proposed is novel and can indeed have a strong impact in the field of epitranscriptomics / genomics. Aside from some strengthening of experimental data (particularly computational/informatics data, see below), I feel that there's a huge missed opportunity here to make a substantial conceptual advance. Because what the authors actually show that A3A editing can predict mutation, *in situ*. This is a big deal. Why?

Because while levels of editing here are considered "low", levels of mutation per genome per population doubling are almost always much lower (the authors can try to estimate this: a new mutation might be acquired in one of two alleles making the "rate" per genome minuscule; but a 7%-20% rate among transcripts, for well expressed genes, is not trivial in terms of in vivo preferred catalytic substrate). This means that the transcript editing rate will always be much higher than the mutation rate on the cognate gene. Given that these proteins are anti-inflammatory, one can hypothesise that A3A's main enzymatic function in the cell as an editor first, with "in situ" mutation as collateral damage (of course proof of that rather than correlation, is a different story). Still, conceptually, this turns the whole idea of "off target" or "ectopic" mutation on its head. The authors might consider incorporating this even as an idea, in their study - this is completely up to them.

We would like to thank the reviewer for raising this important point. We agree with the reviewer in that, because up to 20% of the DDOST RNA is mutated, the RNA editing by A3 may occur much more efficiently than DNA mutations, which are quite rare per cell division. Indeed, we did not see any DNA mutation in DDOST in the thousands of tumor samples that we analyzed. However, we would like to point out that only DNA mutations but not RNA mutations can be inherited during cell proliferation, so it is impossible to directly compare the impacts of DNA and RNA mutations. Indeed, the reviewer makes an excellent point that A3A's main function may be "an RNA editor" instead of "a DNA mutator" because it edits RNA transcripts of anti-inflammatory gene, suggesting that RNA editing by A3A may not be "off target" and hinting that this may even be the main function of A3A. We have now included this idea in the discussion. We thank the reviewer for raising this important point.

*note: "low" is not necessarily accurate; could be high editing within a heterogeneous cell population; also note that RNA editing itself imparts substantial heterogeneity - see Paz-Yaacov et al, Cell Reports 2015 among others; finally note that levels of the preeminent RNA editor ADAR1 are notoriously "bimodal" - and yet the "low editing" events can be just as important as the "high editing" ones - see Picardi et al, RNA 2017

This is another very good point. When we see "low" levels of editing (e.g. 3% of total droplets), it could mean that 100% of cells have a small fraction of the RNA edited (e.g. 3% of the RNA), or that a small fraction of cells (e.g. 3% of cells) has 100% of the RNA edited. It is difficult to distinguish these two possibilities because our RNA samples were prepared from cell populations. As pointed out by the reviewer, previous studies have provided evidence that the truth is likely between the two extreme scenarios. We have discussed this point in the discussion.

Specific comments:

Abstract

- "APOBEC proteins ... in cancers.": Not all APOBEC proteins are deaminases and not all of them are driving mutagenesis in cancer (e.g. APOBEC2 and 4). The authors should rewrite this sentence so as to make this clear.

Good point. We have now modified the first sentence of the abstract by: "APOBEC3A is a cytosine deaminase driving mutagenesis in cancers."

Introduction

- "Importantly, A3A-expressing cells become dependent on the ATR..": Define ATR as it is mentioned for the first time

We have now defined ATR (ataxia telangiectasia and Rad3-related protein) in the manuscript.

- "These RNA mutations..." I would recommend that for base changes on the RNA, the authors should call these "RNA editing" events instead of "mutations", so that the standalone "mutation" term is not confused with regards to the nucleic acid it occurs (RNA or DNA). Alternatively they can be termed "base editing" events.

This is an important point, and we agree with the need for terminological clarity. We have now modified the manuscript to make it clear that we consider "RNA editing" and "RNA mutation" as synonymous terms, and we do not use the standalone term "mutation" to refer to events in RNA.

Results

APOBEC3A-signature mutations do not correlate with APOBEC3A expression in tumors

- Figure 1A-B: the visualisation is very well-done, however it is not clear what the x-axis measures. In the corresponding legend, the authors write “plotted by their level of mutations in the TpC motif and their mutation frequency in RTC versus YTC sequences”; but while the mutation frequency is clear (maybe rename the y axis accordingly), the authors should define more clearly what axis x measures, or in other words define what is meant by “level of mutations” . The in-text explanation still does not address this fully.

We apologize that the analysis in the figures was not explained clearly. We have now added an expanded computational methods section detailing exactly how the analyses were performed. In particular, the x-axis shows a directional metric of A3A vs. A3B mutation character, which we explain in the methods section. This metric is based on the published observation (Chan et al. Nature Genetics 2015 PMID 26258849) that A3A and A3B have distinct preferences for the nucleotide at the "minus-2" position (i.e. just before the TpC site). A3A preferentially targets sites in which the minus-2 position is a pyrimidine (C or T), whereas A3B prefers a purine (A or G) at that position. Thus, we calculate A3A mutation character as the fraction of C mutations in the context YTC, and we calculate A3B mutation character as the fraction of C mutations in the context RTC. The directional metric shown on the x-axis of Fig.1AB combines these two metrics by subtracting them, as now described in the methods.

- Figure 1C and 1D: please, rename the y axis titles so that it's clear that it's the mutation frequency.

We have edited the y-axis titles to "APOBEC3A mutation frequency" as suggested.

APOBEC-signature mutations in RNA stem-loops in tumors

- “To investigate whether A3A... DNA from the same patient”: this is a very crucial step in the analysis. Several mutation calling pipelines fail to accurately report the mutations from WGS data. It is important that upon filtering the SNPs from the transcriptome SNVs for reporting the RNA editing events, the authors should make sure that the mutations they have called are real. That can be achieved with tools such as bam-readcount or IGV (manual spot checking). The authors should also provide the detailed bioinformatic analysis in the “APOBEC analysis in patient DNA-Seq and RNA-Seq data” under Materials & Methods. In particular they should report: how the data were mapped, to which reference human genome version, how DNA mutations were called (I recommend to take the union of at least 2 pipelines), how mutations were computationally validate, how RNA SNVs were called, how RNA editing was called, which sites were considered (statistical significance?), how the annotation was performed. And for all of the above all the crucial parameters that differ from the default settings should be reported.

We fully appreciate the reviewer's point about the danger of missing germline DNA polymorphisms (or somatic DNA mutations). This could lead us to falsely nominate RNA editing events. The reviewer is exactly right: previous approaches for identifying RNA editing events have relied on having DNA mutation calls to "subtract" from the putative RNA editing events. This makes the approaches highly dependent on the exact details of how the DNA mutations were called. When we attempted to use this kind of approach, we found that our results were overwhelmed with false positives. Therefore we developed a custom approach that was designed to be extremely conservative. Our goal was to find specific C residues in the transcriptome that are overwhelmingly "wild

type" at the DNA level, across large numbers of people and tumors, but frequently show C->U RNA editing in tumors with active A3A. To do this, we used a custom multi-step filtering approach, which we now describe fully in the computational methods. Our approach was based on "masking" sites from analysis if they showed any of the following deficiencies: we masked sites if they were poorly covered (<14 reads in either the normal sample's or tumor sample's DNA BAM, or in the tumor sample's RNA BAM), or if there was any hint of a polymorphism, mutation, or sequencing noise in the normal or tumor DNA. Furthermore, we leveraged our published "Panel of Normals" approach for further masking sites of common polymorphisms or sequencing noise. Crucially, to remove sites that undergo non-APOBEC-related RNA editing, we used an independent cohort of APOBEC-negative gastric cancer samples to mask any such sites. Finally, once we had thoroughly masked these variable or insufficiently covered sites, then we looked at the amount of RNA editing at each C site in each sample of our main bladder/cervical/head-and-neck cohort. Gratifyingly, the top 50 RNA editing hotspots we found were all much more highly edited in the A3A-High patients of the cohort, compared to A3A-Low patients (new Supplementary Fig. 3 to 6). We selected our top two RNA editing hotspots (in the genes DDOST and CYFIP1) for validation in a larger panel of tumor samples, and in vitro, and in clinical samples.

- Additionally, the authors claim that the editing events or mutations they have called are in RNA or DNA stem loops. It seems that their top candidates (Fig 3B, 3C) are indeed mutated or edited in such loops, but there is barely any computational evidence throughout the manuscript for that. Authors should show that there's an enrichment in such loops (perhaps fisher test compared to random positions) and also include this analysis in the annotation piece mentioned above.

The reviewer raises a good point. We have now conducted a new analysis as suggested. We evaluated the background of random positions across the transcriptome, classifying each site by its potential hairpin strength, using our previously published method (Buisson et al. Science 2019). Our new Supplementary Fig. 7 shows this background distribution in the upper panel. In contrast, our set of top 50 RNA editing hotspots is shifted strongly toward strongly paired hairpins (lower panel). The difference between these two distributions was extremely statistically significant ($p = 1.7 \times 10^{-21}$, t-test). This was an important point to add to the manuscript, and we thank the reviewer for this suggestion. We have updated the computational methods to explain this new analysis.

Detection of APOBEC3A activity in blood cancer samples

- The 18 patients analysed from AML and MPN, are correlating indeed with the assay developed by the authors, demonstrating that the RNA editing activity detected is attributed to A3A activity. However, I recommend that the authors should analyse the cancer patients with low A3A expression as well where there they should see the reverse from what described in the cohort of 18 patients.

We have included a few patient samples in our analysis in which A3A mRNA was not detected by qPCR. In these samples, the levels of A3A mRNA may be too low to be detected by qPCR, or the amounts of total mRNA may be too low for reliable qPCR. Our ddPCR data suggest that A3A activity can be detected even in the patient samples with undetectable A3A mRNA, arguing that ddPCR is a more sensitive assay to detect A3A in tumors. In addition, we also identified patient samples with low A3A expression and low level of RNA editing (Fig. 6B-C). These patient samples could be truly negative for A3A or contain extremely low A3A activity below the detection limit of ddPCR.

REVIEWERS' COMMENTS:

Reviewer #1 (Remarks to the Author):

The authors have modified the manuscript and addressed all points raised by this reviewer and the others.

Reviewer #2 (Remarks to the Author):

The revised version of the manuscript provides not only the revised text but also additional data to support the figures.

In my opinion authors addressed my previous comments successfully, with some small exceptions listed below.

1. It is still not described how the threshold was set in ddPCR. Was it manually or automatically. It is a well known fact that automatic threshold setting in Quantasoft does not work perfectly, therefore this information is important.

2. I'm still not convinced by the answer about normalization ("In contrast to qPCR, ddPCR is a method of direct quantification (absolute quantification) and it does not require an internal reference"), therefore I would ask for additional clarification in the text.

In case authors want to say that normalization is not important because they use duplex assay and each reaction gives a readout of mutated and non-mutated target, this can be considered as self-normalized in a way. Otherwise if we consider only direct absolute quantification you need any kind of normalizer, when you want to compare samples, which might not have the same RNA extraction efficiency.

3. The statement "ddPCR is based on the partitioning of PCR reactions into small droplets, most of which contain a single copy of the template sequence" is still not describing the ddPCR system properly and does not follow Poisson's distribution. Theoretically, one can quantify roughly 1-200,000 targets in one ddPCR reaction with 20,000 droplets. This means that when almost fully saturated, you can have on average more than 9.9 targets per droplet, with almost no negative droplets left. I would suggest rephrasing into something like: "ddPCR is based on the partitioning of PCR reactions into small droplets, following Poisson's distribution, resulting in partitions containing zero, one or more copies of the template sequence". Please modify

your sentence according to this proposal.

David Dobnik

Reviewer #3 (Remarks to the Author):

Buisson and colleagues were relatively responsive to reviewer requests. The manuscript is significantly cleaned-up and should be ready for prime time.

REVIEWERS' COMMENTS:

Reviewer #2 (Remarks to the Author):

The revised version of the manuscript provides not only the revised text but also additional data to support the figures.

In my opinion authors addressed my previous comments successfully, with some small exceptions listed below.

We thank the reviewer for his appreciation of our revision.

1. It is still not described how the threshold was set in ddPCR. Was it manually or automatically. It is a well known fact that automatic threshold setting in Quantasoft does not work perfectly, therefore this information is important.

The threshold was set manually. For each dataset, an excel file was generated to show the number of droplets in each category (empty droplets, droplets containing only WT template, droplets containing only edited template, and droplets containing both WT and mutated templates), as well as the concentration of WT or edited template (copies per microliter) for each of the categories. Only samples with 3 or more HEX fluorescence-positive droplets and more than 0.25 mutated DDOST copies per microliter were classified as positive for A3A activity. We have now modified our statement by: "To distinguish positive signals from background noise and establish a threshold for significant RNA-editing activity, we manually applied a cutoff of three HEX fluorescence-positive droplets and 0.25 mutated DDOST copies per microliter (20,000 droplets were generated for each sample in the initial step of ddPCR)."

2. I'm still not convinced by the answer about normalization ("In contrast to qPCR, ddPCR is a method of direct quantification (absolute quantification) and it does not require an internal reference"), therefore I would ask for additional clarification in the text.

In case authors want to say that normalization is not important because they use duplex assay and each reaction gives a readout of mutated and non-mutated target, this can be considered as self-normalized in a way. Otherwise if we consider only direct absolute quantification you need any kind of normalizer, when you want to compare samples, which might not have the same RNA extraction efficiency.

We agree with the reviewer comments. In all of our ddPCR experiments, we simultaneously determined the copy numbers of WT and edited templates in all samples, and quantified the fractions of RNA that were edited in each sample. In these experiments, WT templates serve as the internal control for edited templates in each sample, making them self-normalized. When we compared the

RNA-editing activity of different samples, we always compared the fractions of edited RNA in these samples, but not the absolute levels of edited RNA. To avoid any confusion, we have now modified our statement to: “In contrast to qPCR, ddPCR is a method of direct quantification of DNA molecules of specific sequences, making it possible to directly determine the copy numbers of wild-type and edited templates in each sample and the efficiency of editing.”

3.The statement “ddPCR is based on the partitioning of PCR reactions into small droplets, most of which contain a single copy of the template sequence” is still not describing the ddPCR system properly and does not follow Poisson's distribution. Theoretically, one can quantify roughly 1-200.000 targets in one ddPCR reaction with 20.000 droplets. This means that when almost fully saturated, you can have on average more than 9.9 targets per droplet, with almost no negative droplets left. I would suggest rephrasing into something like: “ddPCR is based on the partitioning of PCR reactions into small droplets, following Poisson's distribution, resulting in partitions containing zero, one or more copies of the template sequence”. Please modify your sentence according to this proposal.

We agree with the reviewer's concern. We have now modified our statement following reviewer's suggestion by: “ddPCR is based on the partitioning of PCR reactions into small droplets following the Poisson distribution and resulting in partitions containing zero, one or more copies of the template sequence.”

David Dobnik